# POTEC: OFF-POLICY CONTEXTUAL BANDITS FOR LARGE ACTION SPACES VIA POLICY DECOMPOSITION

**Yuta Saito**
Cornell University
ys522@cornell.edu

**Jihan Yao**
University of Washington
jihany2@cs.washington.edu

**Thorsten Joachims**
Cornell University
tj@cs.cornell.edu

## ABSTRACT

We study off-policy learning (OPL) of contextual bandit policies in large discrete action spaces where existing methods – most of which rely crucially on reward-regression models or importance-weighted policy gradients – fail due to excessive bias or variance. To overcome these issues in OPL, we propose a novel *two-stage* algorithm, called *Policy Optimization via Two-Stage Policy Decomposition (POTEC)*. It leverages clustering in the action space and learns two different policies via policy- and regression-based approaches, respectively. In particular, we derive a novel low-variance gradient estimator that enables to learn a first-stage policy for cluster selection efficiently via a policy-based approach. To select a specific action within the cluster sampled by the first-stage policy, POTEC uses a second-stage policy derived from a regression-based approach within each cluster. We show that a local correctness condition, which only requires that the regression model preserves the relative expected reward differences of the actions within each cluster, ensures that our policy-gradient estimator is unbiased and the second-stage policy is optimal. We also show that POTEC provides a strict generalization of policy- and regression-based approaches and their associated assumptions. Comprehensive experiments demonstrate that POTEC provides substantial improvements in OPL effectiveness particularly in large and structured action spaces.

## 1 INTRODUCTION

Many interactive systems (e.g., language models, recommender systems) are increasingly controlled by automated decision-making policies that learn from historical interactions. These interactions consist of the context (e.g., prompt, user profile), the action chosen by the logging policy (e.g., sentence, recommended product), and the resulting reward (e.g., click). Using such logged interactions, a goal is to train a new policy that improves the expected reward. This *off-policy learning* (OPL) task is of great practical relevance, as it enables us to improve system effectiveness without the risky, slow, and potentially unethical use of online exploration (Saito & Joachims, 2021; Gao et al., 2022).

A highly effective approach to OPL is policy learning by estimating the policy gradient, which has resulted in a number of practical OPL methods for small action spaces (Swaminathan & Joachims, 2015a;b; Joachims et al., 2018; Su et al., 2019; 2020a; Metelli et al., 2021). Unfortunately, this policy-based approach can deteriorate dramatically for large action spaces, which are prevalent in many potential applications of OPL where there exist millions of items (e.g., recommendations of movies, songs, products). In particular, in such large-scale environments, existing policy-based methods, which are mostly based on importance-weighted policy gradients, can collapse due to extremely large variance (Saito & Joachims, 2022; Cief et al., 2024; Sachdeva et al., 2024). While regression-based approaches, which learn the expected reward function and choose the action with the highest predicted reward, could potentially circumvent the variance issue, they are known to suffer from high bias due to model misspecification (Farajtabar et al., 2018; Voloshin et al., 2019; Sachdeva et al., 2020; Saito et al., 2021a) and thus do not provide a readily available solution either.

To overcome this bias and variance dilemma of OPL arising particularly in large action spaces, we develop a novel *two-stage* OPL algorithm called *Policy Optimization via Two-Stage Policy Decomposition (POTEC)*. POTEC operates under a novel policy decomposition framework, wherein the typical overall policy (marginal action distribution) is decomposed into first-stage and second-

stage policies via an action cluster space. The first-stage policy identifies promising action clusters (cluster distribution), while the second-stage policy selects the optimal action within a specific cluster sampled from the first-stage policy (conditional action distribution). A key feature of POTEC is its distinct learning approaches for the policy at each stage. The first-stage policy is learned using a policy-based approach with a novel policy gradient estimator, called the POTEC gradient estimator. It relies on importance weighting in the action cluster space to estimate the value of clusters while using a *pairwise* reward model to deal with the effect of individual actions within each cluster. We show that our gradient estimator is unbiased under *local correctness* (Saito et al., 2023), requiring only that the regression model accurately preserves the relative reward differences within each action cluster. We also show that we can leverage the regression model used in the POTEC gradient estimator to construct a second-stage policy through a regression-based approach.

Compared to standard policy-based methods, the POTEC gradient estimator for the first-stage policy exhibits significantly lower variance in large action spaces. This is because it applies importance weighting to only the action cluster space, which is considerably more compact than the original action space. Furthermore, POTEC is more resilient to estimation bias than typical regression-based approaches, since our first-stage policy is based on an unbiased policy gradient and the second-stage policy only needs to learn the relative value differences between actions, which is less demanding than conventional absolute reward regression. Moreover, we show that POTEC and local correctness provide a full spectrum of OPL approaches whose endpoints are policy- and regression-based methods and their associated reward-modeling conditions. Experiments on synthetic and real-world recommendation datasets demonstrate that POTEC can provide substantially more effective OPL than conventional methods particularly when the per-action training data size is small.

**Related Work.** There is a recent line of work on off-policy *evaluation* (OPE) for large action spaces (Saito & Joachims, 2022; Peng et al., 2023; Saito et al., 2023; Sachdeva et al., 2024; Cief et al., 2024; Aouali et al., 2024; Taufiq et al., 2024). In particular, Saito & Joachims (2022) tackle the problem of OPE for large action spaces and propose the marginalized IPS (MIPS) estimator that leverages pre-defined action embeddings. Peng et al. (2023), Sachdeva et al. (2024), and Cief et al. (2024) propose methods to learn such a structure in the action space from logged data and further improve MIPS. Compared to these works, which focus solely on the OPE problem, we focus on the OPL problem in large action spaces, where OPE methods cannot be directly applied.

Among these, we particularly distinguish our contributions from those of Saito et al. (2023), which develop the OffCEM estimator to handle OPE in large action spaces. Although our algorithm is inspired by the *reward function* decomposition in OffCEM, its application to our concept of *policy* decomposition and the associated *two-stage* OPL algorithm are novel. Moreover, we address the off-policy *learning* (OPL) problem and develop a corresponding algorithm, whereas Saito et al. (2023) focus solely on the off-policy *evaluation* problem in their analysis and experiments. Thus, our work is the first to formulate and propose methods specific to OPL for large discrete action spaces and offers several unique contributions from both methodological and empirical perspectives.

There also exist several related methods aimed at improving sample efficiency in large action spaces for reinforcement learning (RL). Chandak et al. (2019) propose a method to learn action representations to enhance the sample efficiency of on-policy RL. However, the setting of Chandak et al. (2019) is not on *offline* policy learning, and thus their proposed method is not considered as a baseline in our paper. Additionally, the supervised representation learning procedure in their work relies on RL-specific structures (i.e., state transitions), making it inapplicable to our contextual bandit setup. Similarly, Gu et al. (2022) study offline RL in large action spaces and propose a method to learn latent representations in the action space. Their method leverages a data-distributional metric to learn action embeddings. This metric is based on the MDP structure, and how to apply it to the offline contextual bandit problem is not discussed, making it incomparable in our experiments.

Finally, Ban & He (2021) and Zhu et al. (2022) study *online* bandits in large action spaces where active exploration is allowed, whereas we focus on the *offline* setup with no additional exploration.

## 2   OFF-POLICY LEARNING FOR CONTEXTUAL BANDITS

We formulate OPL under the general contextual bandit process, where a decision maker repeatedly observes a context $x \in \mathcal{X}$ drawn i.i.d. from an unknown distribution $p(x)$. Given context $x$, a

potentially stochastic *policy* $\pi(a \mid x)$ chooses action $a$ from a finite action space denoted as $\mathcal{A}$. The reward $r \in [0, r_{\max}]$ is then sampled from some unknown conditional distribution $p(r \mid x, a)$. We define the *value* of policy $\pi$ as a measure of its effectiveness:

$$V(\pi) := \mathbb{E}_{p(x)\pi(a|x)p(r|x,a)}[r] = \mathbb{E}_{p(x)\pi(a|x)}[q(x,a)],$$

where we use $q(x, a) := \mathbb{E}[r \mid x, a]$ to denote the expected reward function given $x$ and $a$.

Our goal is to learn a new policy $\pi_\theta$, parameterized by $\theta$, to maximize the policy value as

$$\theta^* = \arg\max_{\theta \in \Theta} V(\pi_\theta).$$

The logged data we can use for performing OPL takes the form $\mathcal{D} := \{(x_i, a_i, r_i)\}_{i=1}^n$, which contains $n$ independent observations drawn from the logging policy $\pi_0$.

Below, we describe two typical approaches to OPL, namely the policy-based and regression-based approaches, and summarize their limitations, particularly in large action spaces.

**The policy-based approach** learns the policy parameter via iterative gradient ascent as $\theta_{t+1} \leftarrow \theta_t + \nabla_\theta V(\pi_\theta)$. Since we do not know the true gradient

$$\nabla_\theta V(\pi_\theta) = \mathbb{E}_{p(x)\pi_\theta(a|x)}[q(x,a)\nabla_\theta \log \pi_\theta(a \mid x)],$$

we need to estimate it from the logged data. A common way to do so is to apply importance weighting:

$$\nabla_\theta \widehat{V}_{\text{IPS}}(\pi_\theta; \mathcal{D}) := \frac{1}{n} \sum_{i=1}^n w(x_i, a_i) r_i s_\theta(x_i, a_i), \tag{1}$$

where $w(x, a) := \pi_\theta(a \mid x)/\pi_0(a \mid x)$ is the vanilla importance weight defined with respect to the action space $\mathcal{A}$ and $s_\theta(x, a) := \nabla_\theta \log \pi_\theta(a \mid x)$ is the policy score function.

Eq. (1) is unbiased (i.e., $\mathbb{E}[\nabla_\theta \widehat{V}_{\text{IPS}}(\pi_\theta; \mathcal{D})] = \nabla_\theta V(\pi_\theta)$) under the following condition.

**Condition 2.1.** (Full Support) The logging policy $\pi_0$ is said to have full support if $\pi_0(a \mid x) > 0, \ \forall (x, a) \in \mathcal{X} \times \mathcal{A}$.

For large action spaces, unfortunately, this requirement of full support is problematic for two reasons. First, violating the requirement can introduce substantial bias (Sachdeva et al., 2020; Felicioni et al., 2022). Second, fulfilling the requirement for large action spaces leads to excessive variance, since $\pi_0(a \mid x)$ becomes extremely small (Saito & Joachims, 2022; Sachdeva et al., 2024). At first glance, *doubly-robust* (DR) estimation may appear helpful for dealing with the variance issue.

$$\nabla_\theta \widehat{V}_{\text{DR}}(\pi_\theta; \mathcal{D}) := \frac{1}{n} \sum_{i=1}^n \left\{ w(x_i, a_i)(r_i - \hat{q}(x_i, a_i))s_\theta(x_i, a_i) + \mathbb{E}_{\pi_\theta(a|x_i)}[\hat{q}(x_i, a)s_\theta(x_i, a)] \right\}. \tag{2}$$

DR uses a reward function estimator $\hat{q}(x, a)$ while maintaining unbiasedness under Condition 2.1, and its variance is often lower than that of Eq. (1). However, unless the rewards are close to deterministic and the reward estimates $\hat{q}(x, a)$ are close to perfect, its variance can still be extremely large due to importance weighting in the action space, which leads to inefficient OPL in large action spaces (Saito & Joachims, 2022; Peng et al., 2023; Sachdeva et al., 2023). The issue of the IPS and DR policy gradients can be seen by calculating their variance (for a particular parameter $\theta \in \mathbb{R}^d$) as

$$n \operatorname{tr} \left( \operatorname{Cov}_{\mathcal{D}} \left[ \nabla_\theta \widehat{V}_{\text{DR}}(\pi_\theta; \mathcal{D}) \right] \right) = \sum_{j=1}^d \left\{ \mathbb{E}_{p(x)\pi_0(a|x)}[(w(x,a)s_\theta^{(j)}(x,a))^2 \sigma^2(x,a)] \right.$$
$$\left. + \mathbb{E}_{p(x)} \left[ \mathbb{V}_{\pi_0(a|x)}[w(x,a)\Delta_{q,\hat{q}}(x,a)s_\theta^{(j)}(x,a)] \right] + \mathbb{V}_{p(x)} \left[ \mathbb{E}_{\pi(a|x)}[q(x,a)s_\theta^{(j)}(x,a)] \right] \right\}, \tag{3}$$

where $\sigma^2(x, a) := \mathbb{V}[r \mid x, a]$ and $\Delta_{q,\hat{q}}(x, a) := q(x, a) - \hat{q}(x, a)$. $s_\theta^{(j)}(x, a)$ is the $j$-th dimension of the score function. Note that the variance of IPS can be obtained by setting $\hat{q}(x, a) = 0$. The variance reduction of DR comes from the second term where $\Delta_{q,\hat{q}}(x, a)$ is smaller than $q(x, a)$ if $\hat{q}(x, a)$ is accurate. However, we can also see that the variance contributed by the first term can be extremely

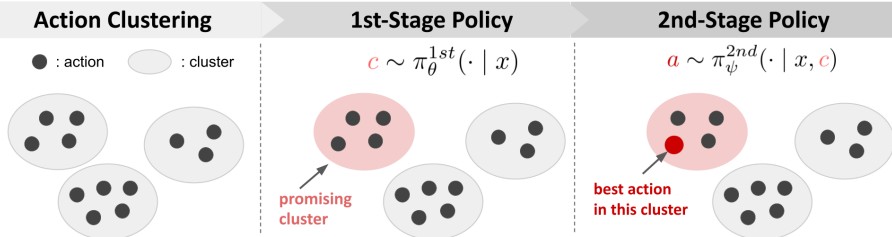

Figure 1: **The Two-Stage Off-Policy Learning Procedure of Our POTEC Algorithm**, which first forms action clustering $c_a$, and then identifies a promising cluster by the 1st-stage policy $\pi_\theta^{1st}$, and finally picks the best action in the cluster by the 2nd-stage policy $\pi_\psi^{2nd}$.

large, regardless of the accuracy of $\hat{q}(x, a)$, when the reward is noisy and the weights $w(x, a)$ become large, which occurs when $\pi_\theta$ assigns large probabilities to actions that are less likely under $\pi_0$.

**The regression-based approach** employs an off-the-shelf supervised machine learning method, such as neural networks, to estimate the reward function $q(x, a)$, for example, by solving

$$\theta = \arg\min_\theta \sum_{(x,a,r)\in\mathcal{D}} \ell\big(r, \hat{q}_\theta(x, a)\big),$$

for some loss function $\ell$ such as the squared loss. Then, it transforms the estimated reward function $\hat{q}_\theta(x, a)$ into a decision-making rule $\pi_\theta(a \mid x)$. This approach avoids the use of importance weighting and is therefore relatively robust to high variance compared to the policy-based approach, even in large action spaces (Jeunen & Goethals, 2021). However, it is widely acknowledged that this approach may fail significantly due to bias issues resulting from the difficulty in accurately estimating the expected reward for every action in $\mathcal{A}$ (Farajtabar et al., 2018; Voloshin et al., 2019).

## 3 THE POTEC ALGORITHM

The following proposes a new OPL algorithm, named **POTEC**, that circumvents the challenges of policy-based and regression-based approaches for large action spaces. As depicted in Figure 1, POTEC leverages the following novel decomposition of an *overall* policy.

$$\pi_{\theta,\psi}^{overall}(a \mid x) = \sum_{c\in\mathcal{C}} \pi_\theta^{1st}(c \mid x)\pi_\psi^{2nd}(a \mid x, c), \tag{4}$$

where the marginal action-selection (overall) policy ($\pi_{\theta,\psi}^{overall}$) is decomposed into the cluster-selection (first-stage) policy ($\pi_\theta^{1st}$) and conditional action-selection (second-stage) policy ($\pi_\psi^{2nd}$), parametrized by $\theta$ and $\psi$, respectively. This policy decomposition is defined via some pre-defined clustering structure in the action space, where $c_a \in \mathcal{C}$ represents the cluster to which action $a$ belongs (typically $|\mathcal{C}| \ll |\mathcal{A}|$). Action clusters are learnable, by applying clustering algorithms, such as KMeans, in the action feature space or to the averaged estimated rewards for each action, $\hat{q}(a) := (1/n)\sum_{i=1}^n \hat{q}(x_i, a)$, as the embedding of $a$, which is a heuristic but performed effectively in our experiments. Note that although we consider context-independent and deterministic action clusters in the main text, our framework can be extended to more general types of action clustering (i.e., context-dependent and stochastic).

Leveraging this decomposition, POTEC **(i)** trains the 1st-stage policy $\pi_\theta^{1st}$, a parameterized distribution over the cluster space $\mathcal{C}$, using a policy-based approach, and **(ii)** trains the 2nd-stage policy $\pi_\psi^{2nd}$, a parameterized distribution over the action space $\mathcal{A}$ conditional on a cluster sampled by the 1st-stage policy, using a regression-based approach. The underlying intuition is that we can apply a policy-based approach to identify promising action clusters with low bias and variance, as the cluster space is much smaller than the original action space. We then apply a regression-based 2nd-stage policy to identify promising actions within each cluster, minimizing variance. The resulting overall policy is more robust to reward modeling errors than typical regression-based approaches because we apply the regression-based policy only within each cluster.[1]

---

[1]One might wonder why not use a regression-based policy for the first stage and a policy-based approach for the second stage. While this idea may also seem reasonable at first, to produce an unbiased policy gradient

When performing inference for an incoming context $x$, we first sample a cluster from the 1st-stage policy as $c \sim \pi_\theta^{1st}(\cdot \,|\, x)$. We then apply the 2nd-stage policy to choose the action in the cluster as $a \sim \pi_\psi^{2nd}(\cdot \,|\, x, c)$. This procedure is equivalent to producing an action as $a \sim \pi_{\theta,\psi}^{overall}(\cdot \,|\, x)$.

The following describes how to train 1st- and 2nd-stage policies to improve the overall policy.

## 3.1 Training the 1st-Stage Policy $\pi_\theta^{1st}$

First, we develop a training procedure for the 1st-stage policy given a (pre-trained) 2nd-stage policy. Then, the theoretical analysis of the proposed training procedure will naturally tell us how we should construct the 2nd-stage policy (which will be described in the next subsection).

As mentioned earlier, given a (pre-trained) 2nd-stage policy $\pi_\psi^{2nd}$, we consider training the 1st-stage policy $\pi_\theta^{1st}$, parameterized by $\theta$, via a policy-based approach as below.

$$\theta_{t+1} \leftarrow \theta_t + \nabla_\theta V(\pi_{\theta,\psi}^{overall}) \tag{5}$$

This performs gradient ascent of $\theta$ with the aim of improving the value of the overall policy $\pi_{\theta,\psi}^{overall}$. The true policy gradient in Eq. (5) is given as follows (derived in Appendix D),

$$\nabla_\theta V(\pi_{\theta,\psi}^{overall}) = \mathbb{E}_{p(x)\pi_\theta^{1st}(c|x)} \left[ q^{\pi_\psi^{2nd}}(x,c) s_\theta(x,c) \right], \tag{6}$$

where we use $q^{\pi_\psi^{2nd}}(x,c) := \mathbb{E}_{\pi_\psi^{2nd}(a|x,c)}[q(x,a)]$ to denote the value of cluster $c$ under a 2nd-stage policy[2] and $s_\theta(x,c) := \nabla_\theta \log \pi_\theta^{1st}(c \,|\, x)$ to denote the policy score function of the 1st-stage policy.

Hence, given a 2nd-stage policy, our objective is to estimate the policy gradient in Eq. (6) to train a 1st-stage policy. We achieve this via the following **POTEC gradient estimator**,

$$\nabla_\theta \widehat{V}_{\text{POTEC}}(\pi_{\theta,\psi}^{overall}; \mathcal{D}) \tag{7}$$

$$:= \frac{1}{n} \sum_{i=1}^n \left\{ w(x_i, c_{a_i})(r_i - \hat{f}(x_i, a_i)) s_\theta(x_i, c_{a_i}) + \mathbb{E}_{\pi_\theta^{1st}(c|x_i)}[\hat{f}^{\pi_\psi^{2nd}}(x_i, c) s_\theta(x_i, c)] \right\},$$

where $w(x,c) := \pi_\theta^{1st}(c \,|\, x)/\pi_0(c \,|\, x)$ is the *cluster importance weight*. Specifically, the first term of Eq. (7) estimates the value of cluster $c$ via cluster importance weighting and the second term deals with the value of individual actions via the regression model $\hat{f}$. Since our policy gradient estimator applies importance weighting with respect to only the action cluster space, it provides a substantial reduction in variance compared to typical estimators such as IPS and DR. We will discuss how we should optimize the regression model $\hat{f}$ based on the following analysis of our gradient estimator.

First, we characterize the bias of the POTEC gradient estimator under the following full cluster support condition (which is less restrictive than Condition 2.1).

**Condition 3.1.** (Full Cluster Support) The logging policy $\pi_0$ has full cluster support if $\pi_0(c \,|\, x) > 0$, $\forall (x,c) \in \mathcal{X} \times \mathcal{C}$.

In the following theorem, we denote with $\Delta_q(x, a, b) := q(x,a) - q(x,b)$ the difference in the expected rewards between the pair of actions $a$ and $b$ given $x$, which we call the **relative value difference** of the actions. $\Delta_{\hat{f}}(x, a, b) := \hat{f}(x,a) - \hat{f}(x,b)$ is an estimate of the relative value difference between $a$ and $b$ based on a regression model $\hat{f}$.

**Theorem 3.2.** *(Bias Analysis) If Condition 3.1 is true, the POTEC gradient estimator in Eq. (7) has the following bias for some given regression model $\hat{f}(x, a)$,*

$$\text{Bias}(\nabla_\theta \widehat{V}_{\text{POTEC}}(\pi_{\theta,\psi}^{overall}; \mathcal{D})) = \mathbb{E}_{p(x)\pi_0^{1st}(c|x)} \Big[ \sum_{a<b:c_a=c_b=c} \pi_0^{2nd}(a \,|\, x, c) \pi_0^{2nd}(b \,|\, x, c)$$

$$\left( \Delta_q(x,a,b) - \Delta_{\hat{f}}(x,a,b) \right) \left( w(x,b) - w(x,a) \right) s_\theta(x,c) \Big], \tag{8}$$

---

in the second stage using a policy-based approach, we would need to apply the typical importance weight $w(x,a)$, resulting in the same variance issues as typical methods. Hence, we focus on the proposed approach of using a policy-based method in the first stage and a regression-based approach in the second stage, leading to a substantial variance reduction while performing unbiased gradient estimation.

[2]This implies that the optimal cluster that should be chosen by the 1st-stage policy can be different given different 2nd-stage policies. Appendix D.1 elaborates on this via a numerical example.

where $a, b \in \mathcal{A}$.

The proof is given in Appendix D.2. Theorem 3.2 shows that the bias of the POTEC gradient estimator is characterized by the *accuracy of the regression model $\hat{f}$ with respect to the relative value difference*, which is quantified by $\Delta_q(x, a, b) - \Delta_{\hat{f}}(x, a, b)$. When $\hat{f}$ preserves the relative value difference of the actions within each cluster accurately, the second factor in Eq. (8) becomes small and so does the bias of the POTEC gradient estimator. This also suggests that, in an ideal case when the *local correctness* condition (Saito et al., 2023) holds true, the POTEC gradient estimator becomes unbiased ($\mathbb{E}_{\mathcal{D}}[\nabla_\theta \widehat{V}_{\mathrm{POTEC}}(\pi_{\theta,\psi}^{overall}; \mathcal{D})] = \nabla_\theta V(\pi_{\theta,\psi}^{overall})$).

**Condition 3.3.** (Local Correctness) A regression model and action clustering satisfy local correctness if $\Delta_q(x, a, b) = \Delta_{\hat{f}}(x, a, b)$ for all $x \in \mathcal{X}$ and $a, b \in \mathcal{A}$ s.t. $c_a = c_b$.

This implies that, to directly minimize the bias of the POTEC gradient estimator, we should ideally optimize the regression model $\hat{f}_\psi : \mathcal{X} \times \mathcal{A} \to \mathbb{R}$ via the following pairwise regression procedure (Saito et al., 2023) so that we can achieve a small $|\Delta_q(x, a, b) - \Delta_{\hat{f}}(x, a, b)|$ by preserving the relative value difference of the actions within each cluster.

$$\min_{\psi} \sum_{(x,a,b,r_a,r_b) \in \mathcal{D}_{pair}} \ell\left(r_a - r_b, \hat{f}_\psi(x, a) - \hat{f}_\psi(x, b)\right), \tag{9}$$

where $\mathcal{D}_{pair}$ is a dataset augmented for performing pairwise regression, which is defined as $\mathcal{D}_{pair} := \left\{(x, a, b, r_a, r_b) \mid \begin{array}{c} (x_a, a, r_a), (x_b, b, r_b) \in \mathcal{D} \\ x = x_a = x_b, c_a = c_b \end{array}\right\}$. As suggested in Theorem 3.2, $\hat{f}_\psi(x, a)$ characterizes the bias of the POTEC gradient estimator. Even if Eq. (9) is infeasible due to insufficient pairwise data, we can still perform a conventional regression for the expected absolute reward to directly optimize the parameterized function $\hat{f}_\psi(x, a)$ via $\min_\psi \sum_{(x,a,r) \in \mathcal{D}} \ell(r, \hat{f}_\psi(x, a))$ and then use $\hat{f}_\psi$ in Eq. (7). Even for such a conventionally trained regression model $\hat{f}_\psi$, POTEC still has advantages over existing policy gradient estimators, such as IPS and DR, due to its substantially reduced variance as demonstrated in our experiments.

Next, the following shows the variance of the POTEC gradient estimator, and its variance reduction compared to typical importance weighting in the action space.

**Proposition 3.4.** *(Variance Analysis) Under Conditions 3.1 and 3.3, for a particular parameter $\theta \in \mathbb{R}^d$, the POTEC gradient estimator has the following variance.*

$$n\,\mathrm{tr}\left(\mathrm{Cov}_{\mathcal{D}}\left[\nabla_\theta \widehat{V}_{\mathrm{POTEC}}(\pi_{\theta,\psi}^{overall}; \mathcal{D})\right]\right) = \sum_{j=1}^{d} \left\{ \mathbb{E}_{p(x)\pi_0(a|x)}\left[(w(x, c_a)s_\theta^{(j)}(x, c_a))^2 \sigma^2(x, a)\right]\right.$$

$$\left. + \mathbb{E}_{p(x)}\left[\mathbb{V}_{\pi_0(a|x)}\left[w(x, c_a)\Delta_{q,\hat{f}}(x, a)s_\theta^{(j)}(x, c_a)\right]\right] + \mathbb{V}_{p(x)}\left[\mathbb{E}_{\pi_\theta^{1st}(c|x)}\left[q^{\pi_\psi^{2nd}}(x, c)s_\theta^{(j)}(x, c)\right]\right] \right\},$$

*where $\Delta_{q,\hat{f}}(x, a) := q(x, a) - \hat{f}(x, a)$ is the error of $\hat{f}(x, a)$ against $q(x, a)$.*

**Proposition 3.5.** *(Variance Reduction) The difference in the variance of the cluster and vanilla importance weights can be represented as follows.*

$$\mathbb{V}_{p(x)\pi_0(a|x)}[w(x, a)] - \mathbb{V}_{p(x)\pi_0(a|x)\pi(c|x,a)}[w(x, c)] = \mathbb{E}_{p(x)\pi_0(c|x)}\left[\mathbb{V}_{\pi_0(a|x,c)}[w(x, a)]\right]$$

Proposition 3.4 shows that the variance of the POTEC gradient estimator depends only on $w(x, c)$ rather than $w(x, a)$, implying reduced variance compared to IPS and DR. In addition, Proposition 3.5 characterizes the reduction in variance provided by cluster importance weighting of POTEC. It is worth noting that the variance reduction is characterized by the variance of the vanilla importance weight $\mathbb{E}_{p(x)\pi_0(c|x)}\left[\mathbb{V}_{\pi_0(a|x,c)}[w(x, a)]\right]$, which suggests that cluster importance weighting provides increasingly larger variance reduction when typical weighting has a larger variance.

## 3.2 TRAINING THE 2ND-STAGE POLICY $\pi_\psi^{2nd}$

We have thus far developed a policy-based approach for learning an effective cluster selection (1st-stage) policy via the POTEC gradient estimator. The remaining objective is to identify the optimal

Figure 2: The POTEC algorithm and local correctness condition generalize policy- and regression-based approaches and their respective conditions about the reward function $(q(x,a))$ estimation.

---

**Algorithm 1** The POTEC Algorithm

---

**Input:** logged bandit data $\mathcal{D}$, conventionally trained regression model $\hat{q}(x,a)$.
**Output:** 1st-stage (policy-based) policy $\pi_\theta^{1st}$ and 2nd-stage (regression-based) policy $\pi_\psi^{2nd}$

1: Perform action clustering by applying a clustering algorithm (such as KMeans) to the averaged estimated rewards for each action, $\hat{q}(a) := (1/n)\sum_{i=1}^{n} \hat{q}(x_i, a)$, as the embedding of $a$.
2: Perform pairwise regression and obtain $\hat{f}_\psi(x,a)$ as in Eq. (9), which works as the 2nd-stage policy as in Eq. (10) and also as a regression model to help train the 1st-stage policy via the POTEC gradient estimator
3: Perform policy-based learning of the 1st-stage policy based on the POTEC estimator in Eq. (7)

---

actions, given a cluster selected by the 1st-stage policy. In essence, we can simply use the pairwise regression model $\hat{f}_\psi$ from the previous section to establish the 2nd-stage policy $\pi_\psi^{2nd}$, because $\hat{f}_\psi$ is already optimized towards estimating the relative value differences of actions within each action cluster (i.e., local correctness). Specifically, we can construct a 2nd-stage policy based on $\hat{f}_\psi$ as

$$\pi_\psi^{2nd}(a \mid x, c) := \begin{cases} 1 & (a = \arg\max_{a':c_{a'}=c} \hat{f}_\psi(x, a')) \\ 0 & (\text{otherwise}) \end{cases} \quad (10)$$

which implies that the 2nd-stage policy selects the action with the highest value of the pairwise regression function $\hat{f}_\psi$ within the already sampled cluster $c$. This action selection procedure is justified since we have learned the function $\hat{f}_\psi$ so that it can estimate the relative value difference of the actions given a cluster in bias minimization (Eq. (9)). In an ideal scenario where Condition 3.3 holds true, our 2nd-stage policy achieves optimal action selection. In our experiments, we will demonstrate that our overall policy $\pi_{\theta,\psi}^{overall}$ outperforms existing approaches by a considerable margin even with a learned 2nd-stage policy that may not perfectly satisfy local correctness.

### 3.3 THE OVERALL POTEC ALGORITHM AND ITS INTERPRETATION

Algorithm 1 describes the overall procedure of POTEC. It first performs action clustering based on an estimated reward function $\hat{q}$. It then performs pairwise regression (if feasible) and obtains the regression function $\hat{f}_\psi$, which forms the 2nd-stage policy (as in Eq. (10)). Note that if pairwise regression is not feasible, we can instead use the conventional regressor as $\hat{f} \leftarrow \hat{q}$. Then, we train the 1st-stage policy $\pi_\theta^{1st}$ based on the POTEC gradient estimator, which is based on cluster importance weighting and a learned regression model $\hat{f}_\psi(x,a)$, which governs the bias of the estimator.

It is worth mentioning that POTEC and its associated local correctness condition generalize typical OPL approaches, i.e., policy-based and regression-based, as depicted in Figure 2. That is, when there is only one action cluster ($|\mathcal{C}| = 1$), the 2nd-stage policy of POTEC needs to choose the best action in the entire action space, which can be seen as a reduction to the regression-based approach. Moreover, in this case, the local correctness condition becomes relatively stringent (since all actions are grouped into the same cluster), which is akin to the typical condition of the regression-based approach, i.e., globally accurate estimation of the reward function. On the other hand, when the cluster space is equivalent to the original action space ($\mathcal{C} = \mathcal{A}$), the 1st-stage policy selects an action from the original action space, akin to the policy-based approach. In this scenario, local correctness imposes no specific requirements, as each action cluster contains only one unique action. This absence of requirements

aligns with the policy-based approach, which does not necessitate specific conditions for reward function estimation to produce an unbiased gradient. Thus, POTEC and local correctness encompass the full spectrum of existing OPL approaches and respective reward-modeling conditions (Figure 2). As a strict generalization, POTEC generally outperforms both approaches with a good (even if not perfect) selection of the number of clusters, as the following section empirically demonstrates.

It is also worth noting that the number of clusters and clustering algorithm are hyperparameters of POTEC, so we can tune them via the cross-validation procedure. However, it is important to perform careful OPE on validation data to evaluate the performance of each hyperparameter with low bias. To address the variance issue in large action spaces during validation, we would recommend using OPE methods from Sachdeva et al. (2024); Saito et al. (2023); Peng et al. (2023).

## 4 EMPIRICAL EVALUATION

We first evaluate POTEC on synthetic data with ground-truth cluster information to compare the effectiveness of POTEC w/ and w/o true cluster information and w/ and w/o pairwise regression. We then assess the real-world applicability of POTEC on a public recommendation dataset.

### 4.1 SYNTHETIC EXPERIMENT

First, we create synthetic datasets to compare the policy learning algorithms based on their ground-truth policy values. Specifically, we first sample 5-dimensional context vectors $x$ and action features from the standard normal distribution. We then form ground-truth action clusters based on the action features. We synthesize the expected reward function as $q(x, a) = g(x, c_a) + h_{c_a}(x, a)$, where $g(x, c)$ and $h_c(x, a)$ define the values of the cluster and individual actions, respectively, as detailed in Appendix E. We then sample an action $a$ based on the logging policy $\pi_0$, which is defined as below.

$$\pi_0(a|x) := \frac{\exp(\beta \cdot (q(x, a) + \eta_{x,a}))}{\sum_{a' \in \mathcal{A}} \exp(\beta \cdot (q(x, a') + \eta_{x,a'}))}, \quad (11)$$

where $\beta$ is set to 5 as default and the noise $\eta_{x,a}$ is sampled from a normal distribution.

After sampling action $a$ from the logging policy, we finally sample the reward $r$ from a normal distribution with mean $q(x, a)$ and standard deviation $\sigma_r = 1.0$. Repeating the above procedure $n$ times generate logged training data of the form $\mathcal{D} = \{(x_i, a_i, r_i)\}_{i=1}^{n}$.

**Baselines:** We compare POTEC with the regression-based method (Reg-based), IPS-PG (Eq. (1)), and DR-PG (Eq. (2)). We use a neural network with 3 hidden layers to parameterize the policy $\pi_\theta$, $\hat{q}(x, a)$ for DR-PG and Reg-based, and $\hat{f}_\psi$ for POTEC. We apply the variance reduction technique proposed by Lopez et al. (2021) to IPS-PG and DR-PG. In the synthetic experiment, we compare POTEC w/ true clusters that define the true reward function $q(x, a)$ and POTEC w/ learned clusters obtained by performing KMeans to the averaged estimated rewards for each action.[3]

**Results:** Figure 3 shows the policy values (normalized by the value of the logging policy $V(\pi_0)$) of the OPL methods on test data obtained from 30 simulations with varying seeds. Note that we employ default experiment parameters of $n = 4,000$, $|\mathcal{A}| = 500$, and $|\mathcal{C}| = 30$, and the shaded regions in the plots represent 95% confidence intervals of the test policy values estimated via bootstrap.

First, in most situations, POTEC provides significant improvements in policy value over the baseline methods. Specifically, in Figure 3 (i), we observe that POTEC performs increasingly better with larger sample sizes, while the baseline methods remain inferior compared to POTEC. We also see that the policy value of POTEC is particularly higher than the baseline methods when the training data size is small, such as $n \leq 2,000$, which suggests that POTEC is efficient even with a small per-action sample size (i.e., $n/|\mathcal{A}|$), while the baseline methods require even more data for each action to be effective. It is also interesting to compare the policy value of POTEC (w/ true clusters) and POTEC (w/ learned clusters), where we observe that POTEC (w/ true clusters) consistently performs best, while POTEC (w/ learned clusters) achieves competitive performance with POTEC (w/ true clusters) and outperforms the baselines by a large margin across a range of training data sizes. This suggests

---

[3]The number of learned action clusters (i.e., the parameter $K$ of KMeans) is set equal to $|\mathcal{C}|$.

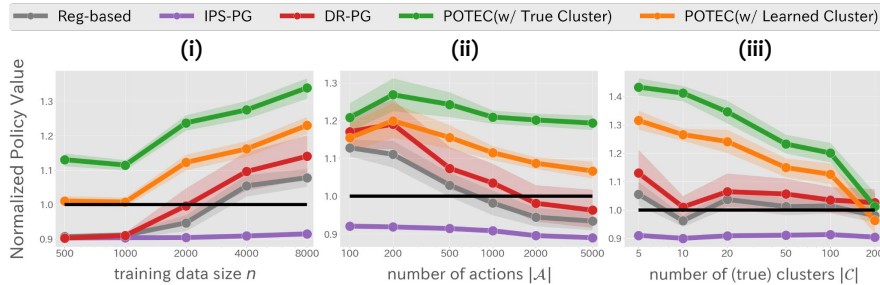

Figure 3: Comparing the test policy value (normalized by $V(\pi_0)$) of the OPL methods, with varying **(i)** training data sizes, **(ii)** numbers of actions, and **(iii)** numbers of (true) clusters, on **Synthetic Data**.

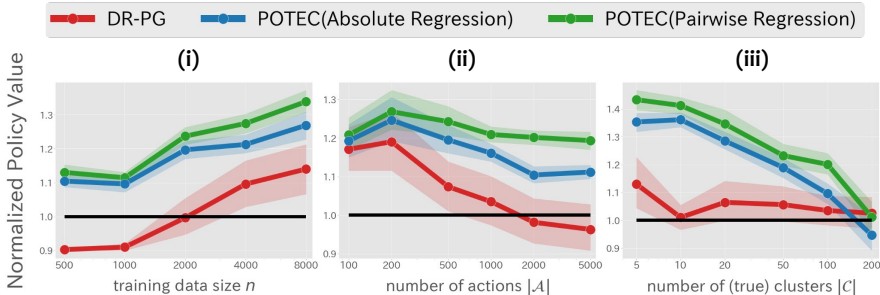

Figure 4: Comparing the test policy value of POTEC w/ and w/o pairwise regression, with varying **(i)** training data sizes, **(ii)** numbers of actions, and **(iii)** numbers of (true) clusters, on **Synthetic Data**.

the real-world applicability of POTEC, even with the feasible and simple clustering procedure based on $\hat{q}(x,a)$. Next, in Figure 3 (ii), we vary the number of actions ($|\mathcal{A}|$) to investigate the robustness of the methods to increasing action spaces. We observe that POTEC performs consistently well, even as the action space grows, as long as the underlying cluster space $\mathcal{C}$ does not expand. In contrast, the performance of the baseline methods clearly deteriorates as the number of actions increases, and they perform even worse than the logging policy when $|\mathcal{A}| \geq 2,000$. We can also see that POTEC (w/ learned clusters) performs substantially better than the baseline methods, particularly when the action space becomes large, even with heuristically learned action clusters. Finally, Figure 3 (iii) evaluates POTEC as we increase the number of (true) clusters while keeping the number of actions fixed. The figure shows that the advantage of POTEC is greatest when the cluster effect can be captured by a small number of underlying clusters. However, even for data with $|\mathcal{C}| = 200$ clusters, POTEC remains competitive with the baselines.[4]

In Figure 4, we present the results of an ablation study for (i) varying training data sizes, (ii) varying numbers of actions, and (iii) varying numbers of (true) clusters, where we compare POTEC with pairwise regression $\hat{f}(x,a)$, as described in Eq. (9), and with the standard absolute regression $\hat{q}(x,a)$. We also include DR-PG, the best method among the baselines, as a reference. From the figure, it is clear that both versions of POTEC significantly outperform the best baseline (DR-PG) in most experimental settings, which empirically highlights the advantage of using POTEC, even with a conventionally trained regression model $\hat{q}(x,a)$. However, it is also evident that performing pairwise regression consistently improves POTEC, making it beneficial to apply when feasible.

## 4.2 REAL-WORLD EXPERIMENT ON KUAIREC

To assess the real-world applicability of POTEC, we now evaluate it on the KuaiRec dataset (Gao et al., 2022), a publicly available recommendation dataset collected on a short video platform, where 1,411 users have viewed all 3,317 videos and provide watch ratio (play duration divided by the video duration) as reward feedback. This *full feedback* nature of the dataset enables an OPL experiment without synthesizing the reward function $q(x,a)$.

---

[4]Note that this experiment varies the ground-truth *underlying* cluster structure, and as a result, the policy values of the baseline methods may not remain constant.

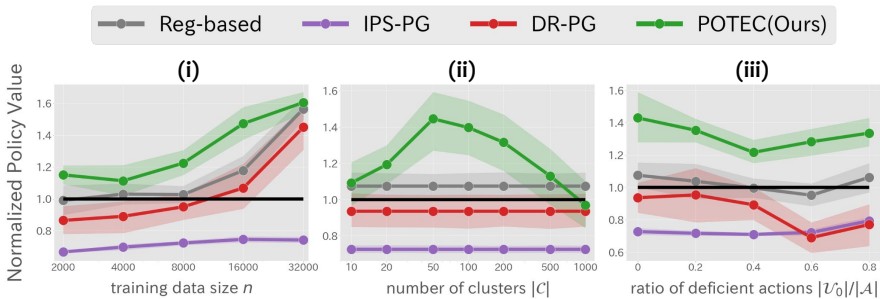

Figure 5: Comparing the test policy value (normalized by $V(\pi_0)$), with varying **(i)** training data sizes, **(ii)** numbers of (learned) clusters, and **(iii)** ratios of deficient actions, on the **KuaiRec** dataset.

To perform an OPL experiment and generate logged training data, we first sample user index $u$ from a uniform distribution, and for each user, we sample an action $a$ based on the logging policy $\pi_0$ defined in Eq. (11). After sampling action $a$, we sample the reward $r$ from a normal distribution with mean $q(x_u, a)$ and standard deviation $\sigma_r = 1.0$ where the expected reward $q(x_u, a)$ is the watch ratio recorded in the dataset. Repeating the above procedure $n$ times generates $\mathcal{D} = \{(x_{u_i}, a_i, r_i)\}_{i=1}^{n}$.

**Results:** We evaluate POTEC against IPS-PG, DR-PG, and Reg-based methods under varying training data sizes $n$, numbers of (learned) clusters $|\mathcal{C}|$, and ratios of deficient actions $|\mathcal{U}_0| := \{a \in \mathcal{A} \mid \pi_0(a|x) = 0\}|/|\mathcal{A}|$ (Sachdeva et al., 2020). For POTEC, we perform a pairwise reward regression using Random Forest to obtain $\hat{f}(x, a)$. Note that, in the real-data, there is no ground-truth action clusters, so we implement only a feasible version of POTEC with learned action clusters.

Figure 5 presents the real-world experiment results for various configurations. First, Figure 5 (i) compares the methods across different training data sizes, showing that POTEC is the most sample-efficient and performs significantly better than the baselines, particularly when the training dataset is small. Second, Figure 5 (ii) compares POTEC across varying numbers of clusters $|\mathcal{C}|$, a key hyperparameter of POTEC. The figure shows that POTEC is most sample-efficient with a moderate number of clusters, such as $|\mathcal{C}| = 50$ or $|\mathcal{C}| = 100$. It is also noteworthy that POTEC does not underperform the baseline or logging policies even in the worst cases, with overly small ($|\mathcal{C}| = 10$) or large ($|\mathcal{C}| = 1,000$) cluster spaces. Finally, Figure 5 (iii) compares the methods as we vary the violations of the full support condition (Condition 2.1), where we gradually expand the set of *deficient actions* $\mathcal{U}_0$. When the logging policy $\pi_0$ becomes nearly deterministic and offers limited exploration, $|\mathcal{U}_0|$ grows, making OPL more challenging. Indeed, in Figure 5 (iii) we observe that the baseline policy-based methods, particularly DR-PG, struggle as the number of deficient actions $|\mathcal{U}_0|$ increases, which produces greater bias in policy gradient estimation. However, POTEC continues to perform significantly better than the baselines and logging policy, even with many deficient actions, because the POTEC gradient estimator requires only full cluster support (Condition 3.1), which is milder than full support and thus results in smaller bias in the gradient estimation of the 1st-stage policy.

## 5 CONCLUSION AND FUTURE WORK

This work introduces a novel two-stage OPL procedure called POTEC, which is particularly advantageous in large action spaces. POTEC learns the first-stage cluster-selection policy via a new policy gradient estimator that is unbiased under local correctness and has substantially lower variance. The second-stage action-selection policy is learned through pairwise reward regression within each cluster, offering greater robustness to estimation bias compared to traditional regression-based approaches.

Our findings suggest valuable directions for future research. For instance, even though we have empirically demonstrated that POTEC generally outperforms existing OPL methods, we now rely on a heuristic action clustering, which could be seen as a limitation. It would thus be valuable to explore a more principled method, such as a context-dependent and iterative procedure to better satisfy local correctness. Extending POTEC to offline reinforcement learning and large language models beyond contextual bandits is also an intriguing future direction.

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

# A  MORE RELATED WORK

**Off-Policy Evaluation:** Off-policy evaluation of counterfactual policies has recently garnered significant interest in both contextual bandits (Dudík et al., 2014; Farajtabar et al., 2018; Kallus et al., 2021; Kiyohara et al., 2022; 2023; Metelli et al., 2021; Saito & Joachims, 2021; Su et al., 2020a; 2019; Wang et al., 2017) and reinforcement learning (RL) (Jiang & Li, 2016; Kallus & Uehara, 2020; Liu et al., 2018; 2020; Thomas & Brunskill, 2016; Xie et al., 2019). The literature encompasses three main approaches. The first approach, named Direct Method (DM), is defined as:

$$\hat{V}_{\text{DM}}(\pi; \mathcal{D}, \hat{q}) := \frac{1}{n} \sum_{i=1}^{n} \mathbb{E}_{\pi(a|x_i)}[\hat{q}(x_i, a)] = \frac{1}{n} \sum_{i=1}^{n} \sum_{a \in \mathcal{A}} \pi(a \mid x_i) \hat{q}(x_i, a),$$

where $\hat{q}(x, a)$ estimates $q(x, a)$ based on logged bandit data. This approach exhibits lower variance than IPS and has been utilized to address violations of full support (Sachdeva et al., 2020), where IPS can be severely biased. However, DM is often vulnerable to reward function misspecification. This issue is problematic, as the extent of misspecification cannot be easily detected and evaluated for real-world data due to non-linearity or partial observability of the environment (Farajtabar et al., 2018; Sachdeva et al., 2020; Voloshin et al., 2019). The second approach is IPS, which estimates the value of $\pi$ by re-weighting the observed rewards as

$$\hat{V}_{\text{IPS}}(\pi; \mathcal{D}) := \frac{1}{n} \sum_{i=1}^{n} \frac{\pi(a_i \mid x_i)}{\pi_0(a_i \mid x_i)} r_i = \frac{1}{n} \sum_{i=1}^{n} w(x_i, a_i) r_i,$$

where $w(x, a) := \pi(a \mid x)/\pi_0(a \mid x)$ is called the *(vanilla) importance weight*. Under some identification assumptions such as no interference, full support, and no unobserved confounders, IPS provides unbiased and consistent estimation of the value of new policies. However, this approach has a critical drawback: it can suffer from high bias and variance in the presence of numerous actions. First, high bias can occur when the logging policy fails to provide full support (Condition 2.1), which is likely in larger action spaces (Sachdeva et al., 2020; Saito & Joachims, 2022). Furthermore, its variance can be particularly excessive for large action spaces, as the importance weights are prone to taking extremely large values. It is possible to apply weight clipping Su et al. (2020a; 2019); Swaminathan & Joachims (2015b) and self-normalization (Swaminathan & Joachims, 2015c) to somewhat alleviate the variance issue, however, they introduce additional bias in return. DR, which is given as follows, is a third approach that can be considered a hybrid of the previous two approaches, achieving lower bias than DM and lower variance than IPS (Dudík et al., 2014; Farajtabar et al., 2018).

$$\hat{V}_{\text{DR}}(\pi; \mathcal{D}, \hat{q}) := \frac{1}{n} \sum_{i=1}^{n} \left\{ w(x_i, a_i)(r_i - \hat{q}(x_i, a_i)) + \mathbb{E}_{\pi(a|x_i)}[\hat{q}(x_i, a)] \right\}$$

Several recent studies have extended DR to further improve its finite sample accuracy (Su et al., 2020a; Wang et al., 2017; Metelli et al., 2021) or its robustness to model misspecification (Farajtabar et al., 2018; Kallus et al., 2021). Although there is a number of extensions of DR in both bandits (as described above) and RL (Jiang & Li, 2016; Kallus & Uehara, 2020; Thomas & Brunskill, 2016), these variants of DR still face the critical variance issue in large action spaces due to the same reasons as IPS Saito & Joachims (2022); Saito et al. (2023).

To address the fundamental issues of typical OPE estimators for large action spaces, (Saito & Joachims, 2022) proposed a new framework and estimator called Marginalized IPS (MIPS). This approach leverages auxiliary information about the actions, called action embeddings or action features, which are available in many potential applications of OPE such as recommender systems, and provide useful structure in the action space. More specifically, MIPS is defined as:

$$\hat{V}_{\text{MIPS}}(\pi; \mathcal{D}) := \frac{1}{n} \sum_{i=1}^{n} \frac{\pi(e_i \mid x_i)}{\pi_0(e_i \mid x_i)} r_i = \frac{1}{n} \sum_{i=1}^{n} w(x_i, e_i) r_i,$$

where the logged dataset $\mathcal{D} = \{(x_i, a_i, e_i, r_i)\}_{i=1}^{n}$ now contains action embeddings for each data point[5] and $w(x, e) := \frac{\pi(e \mid x)}{\pi_0(e \mid x)} = \frac{\sum_a p(e \mid x, a)\pi(a \mid x)}{\sum_a p(e \mid x, a)\pi_0(a \mid x)}$ is the *marginal importance weight*. This weight is defined with respect to the marginal distributions of the action embeddings induced by the

---

[5] $(x, a, e, r) \sim p(x)\pi_0(a \mid x)p(e \mid x, a)p(r \mid x, a, e)$ where $p(e \mid x, a)$ is an action embedding distribution.

target and logging policies. This enhanced weighting scheme results in significantly lower variance compared to IPS and DR in larger action spaces, while maintaining unbiasedness under the no direct effect assumption. This assumption necessitates that the given action embeddings be informative enough to mediate every causal effect of the actions on the rewards (i.e., $a \perp r \mid x, e$). A similar condition regarding the causal structure has been utilized to address the deficient support problem in OPE (Felicioni et al., 2022; Lee et al., 2022; Peng et al., 2023; Sachdeva et al., 2023) and to conduct causal inference of long-term outcomes through short-term proxies (Athey et al., 2020; 2019; Chen & Ritzwoller, 2021). However, MIPS may still exhibit high variance, similarly to IPS, when the provided action embeddings are high-dimensional and fine-grained. Additionally, it may generate substantial bias if the no direct effect condition is violated and action embeddings fail to explain much of the causal effects of the actions. This bias issue is particularly expected when performing action feature selection on high-dimensional action embeddings to reduce variance Su et al. (2020b); Udagawa et al. (2023).

To circumvent the bias-variance dilemma of MIPS, (Saito et al., 2023) proposed a more general formulation and a refined estimator. Specifically, instead of relying on the often demanding no direct effect condition, (Saito et al., 2023) introduced the conjunct effect model (CEM) of the reward function. The CEM is a useful decomposition of the expected reward function into what is called the cluster effect and residual effect. Building on the CEM, we can employ model-free estimation utilizing cluster importance weights to estimate the cluster effect without bias, and apply model-based estimation using the pairwise regression procedure to estimate the residual effect with low variance as

$$\hat{V}_{\text{OffCEM}}(\pi; \mathcal{D}) := \frac{1}{n} \sum_{i=1}^{n} \left\{ w(x_i, c_{a_i})(r_i - \hat{f}(x_i, a_i)) + \mathbb{E}_{\pi(a \mid x_i)}[\hat{f}(x_i, a)] \right\},$$

where $w(x, c) := \frac{\pi(c \mid x)}{\pi_0(c \mid x)} = \frac{\sum_{a \in \mathcal{A}} \mathbb{I}\{c_a = c\} \pi(a \mid x)}{\sum_{a \in \mathcal{A}} \mathbb{I}\{c_a = c\} \pi_0(a \mid x)}$ is referred to as the *cluster importance weight*. The first term of OffCEM estimates the cluster effect through cluster importance weighting, while the second term addresses the residual effect using the regression model $\hat{f}$, which is ideally learned via a two-step procedure similar to POTEC. As a result, the OffCEM estimator is likely to achieve significantly lower variance than IPS, DR, and MIPS in scenarios with many actions or high-dimensional action embeddings, while often reducing the bias of MIPS since OffCEM does not ignore the residual effect. Our OPL algorithm is inspired by this CEM formulation, and suggests training two distinct policies via policy-based (model-free) and regression-based (model-based) approaches, respectively.

**Off-Policy Learning:** The contextual bandit framework has become a popular and practical approach for online learning and decision-making under uncertainty (Lattimore & Szepesvári, 2020), with many efficient algorithms proposed for exploring the (potentially large or even infinite) action spaces Agrawal & Goyal (2013); Li et al. (2010). However, a pressing need exists for an offline procedure that optimizes decision-making without requiring risky and time-consuming active exploration. As a result, there has been significant recent interest in developing efficient off-policy learning methods in the contextual bandit setting Sachdeva et al. (2020); Saito & Joachims (2021). Fortunately, many real-world interactive systems can often leverage logged interaction data to learn an improved policy fully offline, which allows us to safely enhance the performance of the current system Joachims et al. (2018); London & Sandler (2019); Sachdeva et al. (2020); Saito & Joachims (2021); Swaminathan & Joachims (2015a;b).

As already described in Section 2, there are two main families of approaches in OPL: regression-based and policy-based methods. The regression-based approach relies on a reduction to supervised learning, where a regression estimate is trained to predict the rewards from the logged data Jeunen & Goethals (2021); Sachdeva et al. (2020). To derive a policy, the action with the highest predicted reward is chosen deterministically, or a distribution can be formed based on the estimated rewards as well. A drawback of this straightforward approach is the bias that arises from the misspecification of the regression model. On the other hand, the policy-based approach aims to update the parameterized policy $\pi_\theta$ by performing gradient ascent iterations of the form: $\theta_{t+1} \leftarrow \theta_t + \nabla_\theta V(\pi_\theta)$ at each step $t$ during policy learning. Since the true policy gradient $\nabla_\theta V(\pi_\theta) (= \mathbb{E}_{p(x)\pi_\theta(a \mid x)}[q(x, a)\nabla_\theta \log \pi_\theta(a \mid x)])$ is unknown, it must be estimated from the logged data using OPE techniques, such as IPS (Eq. (1)) and DR (Eq. (2)). However, these estimators necessitate the assumption that the logging policy has full support for every policy in the policy space. This assumption is frequently violated in large

action spaces, leading to significant bias in gradient estimation. Moreover, existing policy gradient estimators heavily rely on the vanilla importance weight with respect to the original (potentially large) action space, resulting in critical variance issues and insufficient off-policy learning for large action spaces. One possible approach to address the variance issue in OPE is to apply conservative or imitation regularization Jeunen & Goethals (2021); Liang & Vlassis (2022); Ma et al. (2019); Swaminathan & Joachims (2015b), which penalize policies that diverge from the logging policy. However, in large action spaces, these regularization techniques often yield a policy that is too close to the logging policy. To tackle the challenges associated with OPE in large action spaces, (Lopez et al., 2021) recently proposed the following selective IPS (sIPS) estimator to estimate the policy gradient.

$$\nabla_\theta \widehat{V}_{\text{sIPS}}(\pi_\theta; \mathcal{D}) := \frac{1}{n} \sum_{i=1}^n \frac{\pi_\theta(a_i \mid x_i, a_i \in \Phi(x_i))}{\pi_0(a_i \mid x_i)} r_i \nabla_\theta \log \pi_\theta(a_i \mid x_i), \tag{12}$$

where $\Phi(x) := \{a \in \mathcal{A} \mid q(x, a) > 0\}$ is the set of relevant actions called the action selector. The idea is to reduce the variance in importance weighting by focusing only on relevant actions assuming that there are many irrelevant actions that have (almost) zero expected rewards in real applications. However, we argue that the variance reduction effect of sIPS is often limited, as it still relies on the logging policy in the denominator. Furthermore, a reliable method for identifying the action selector has not yet been provided.

To address the limitations of existing approaches, we utilize the CEM from Saito et al. (2023) and proposed the POTEC algorithm, which is the first OPL framework to unify regression-based and policy-based approaches. This algorithm trains two separate policies using regression-based and policy-based approaches, respectively.[6] In particular, our POTEC algorithm is expected to outperform typical policy- and regression-based approaches in large action spaces. First, we utilize cluster importance weighting when training the 1st-stage policy and a regression-based approach when training the 2nd-stage policy, which should yield significantly lower variance compared to existing policy-based methods that apply importance weighting over the original action space. Furthermore, our algorithm is likely to be more robust to reward function misspecification than the regression-based approach, as it relies on a provably unbiased policy gradient in the 1st-stage and aims to estimate only the relative value difference in the 2nd-stage. This is arguably a simpler task compared to the absolute value regression of the conventional regression-based approach.

Note that in the context of reinforcement learning (RL), there are some related ideas and methods to improve sample-efficiency in large action spaces. For example, Chandak et al. (2019) propose a method to learn action representation to improve sample-efficiency of on-policy RL. However, the focus of Chandak et al. (2019) is not offline policy learning, and thus its proposed method is not considered as a baseline in our paper. In addition, the supervised representation learning procedure of this paper uses the structure specific to RL (i.e., state transition), so it cannot be applied to our contextual bandit setup. In addition, Gu et al. (2022) study offline RL in large action spaces and propose a method to learn latent representation in the action space. However, the proposed method of Gu et al. (2022) leverages the data-distributional metric to learn action embeddings to deal with large action spaces in offline RL, but the metric is based on the MDP structure, and how to apply the method to the offline contextual bandit problem was not discussed and it is non-trivial.

Table 1: Examples of locally correct regression models

| $a$ | $a_0$ | $a_1$ | $a_2$ | $a_3$ |
|---|---|---|---|---|
| $\phi(x_0, a)$ | 0 | | 1 | |
| $q(x_0, a)$ | 4 | 1 | 3 | 2 |
| $\hat{f}_1(x_0, a)$ | 3 | 0 | 1 | 0 |
| $\Delta(x_0, a, b)$ | 3 | | 1 | |

| $a$ | $a_0$ | $a_1$ | $a_2$ | $a_3$ |
|---|---|---|---|---|
| $\phi(x_0, a)$ | 0 | | 1 | |
| $q(x_0, a)$ | 4 | 1 | 3 | 2 |
| $\hat{f}_2(x_0, a)$ | 50 | 47 | -30 | -31 |
| $\Delta(x_0, a, b)$ | 3 | | 1 | |

| $a$ | $a_0$ | $a_1$ | $a_2$ | $a_3$ |
|---|---|---|---|---|
| $\phi(x_0, a)$ | 0 | | 1 | |
| $q(x_0, a)$ | 4 | 1 | 3 | 2 |
| $\hat{f}_3(x_0, a)$ | 4 | 1 | 3 | 2 |
| $\Delta(x_0, a, b)$ | 3 | | 1 | |

---

[6]Note that DR in Eq. (2) should be classified as a policy-based approach since its aim is to accurately estimate the true policy gradient, even though it employs a regression-based reward function estimator for variance reduction from IPS.

## B  EXAMPLES: LOCALLY CORRECT REGRESSION MODELS

This section provides some examples of regression model $\hat{f}$ that satisfies Condition 3.3 (local correctness). Suppose that there is only a single context $\mathcal{X} = \{x_0\}$ and four actions $\mathcal{A} = \{a_0, a_1, a_2, a_3\}$. The expected reward function $q(x, a)$ and clustering function $\phi(x, a)$ are given as follows.

$$q(x_0, a_0) = 4, \ q(x_0, a_1) = 1, \ q(x_0, a_2) = 3, \ q(x_0, a_3) = 2,$$
$$\phi(x_0, a_0) = 0, \ \phi(x_0, a_1) = 0, \ \phi(x_0, a_2) = 1, \ \phi(x_0, a_3) = 1.$$

Then, Table 1 provides three locally correct regression models ($\hat{f}_1$ to $\hat{f}_3$). More specifically, these example models succeed in preserving the relative value difference of the actions within each action cluster ($c = 0$ for $a_0, a_1$ and $c = 1$ for $a_2, a_3$). In fact, we can see that $\Delta_q(x_0, a_0, a_1) = \Delta_{\hat{f}_1}(x_0, a_0, a_1) = \Delta_{\hat{f}_2}(x_0, a_0, a_1) = \Delta_{\hat{f}_3}(x_0, a_0, a_1) = 3$ and $\Delta_q(x_0, a_2, a_3) = \Delta_{\hat{f}_1}(x_0, a_2, a_3) = \Delta_{\hat{f}_2}(x_0, a_2, a_3) = \Delta_{\hat{f}_3}(x_0, a_2, a_3) = 1$ where $\phi(x_0, a_0) = \phi(x_0, a_1)$ and $\phi(x_0, a_2) = \phi(x_0, a_3)$.

## C  GENERALIZATION OF OUR FRAMEWORK AND POTEC ALGORITHM

In this section, we describe the generalization of our framework and algorithm to the situation under the presence of some predefined action representation $\phi : \mathcal{X} \times \mathcal{A} \to \mathcal{E} \subseteq \mathbb{R}^d$, which is often available in practice and can be used to better parameterize the policy. Under the presence of such action representations, we can first generalize the CEM as follows.

$$q(x, a) = \underbrace{g(x, c(x, \Phi(x, a)))}_{\text{cluster effect}} + \underbrace{h(x, \Phi(x, a))}_{\text{residual effect}}, \tag{13}$$

where $c : \mathcal{X} \times \mathcal{E} \to \mathcal{C}$ provides a discretization in the action representation space $\mathcal{E}$. Note also that the residual effect depends on the representation of the action $\Phi(x, a)$ rather than the atomic actions $a$ as in a simpler version presented in the main text.

Leveraging this general version of the CEM in Eq. (13), we can generalize our POTEC gradient estimator in Eq. (7) in the following two ways.

**Implementation Option 1:**  This option trains a parameterized distribution over the action representation space $\mathcal{E}$ as the 1st-stage policy via the following version of the POTEC gradient estimator.

$$\nabla_\theta \widehat{V}_{\text{POTEC}}(\pi_{\theta,\psi}^{overall}; \mathcal{D}) := \frac{1}{n} \sum_{i=1}^{n} \left\{ w(x_i, c_i)(r_i - \hat{f}(x_i, \Phi(x_i, a_i))) \nabla_\theta \log \pi_\theta(\Phi(x_i, a_i) \mid x_i) \right.$$
$$\left. + \mathbb{E}_{e \sim \pi_\theta^{1st}}[\hat{f}^{\pi_\psi^{2nd}}(x_i, c) \nabla_\theta \log \pi_\theta(e \mid x_i)] \right\}, \tag{14}$$

where $c_i = c(x_i, \Phi(x_i, a_i))$, $\hat{f}^{\pi_\psi^{2nd}}(x, c) := \mathbb{E}_{\pi_\psi^{2nd}}[\hat{f}(x, a)]$ and

$$w(x, c) := \frac{\pi_\theta^{1st}(c \mid x)}{\pi_0^{1st}(c \mid x)} = \frac{\int_{e:c(x,e)=c} \pi_\theta^{1st}(e \mid x)}{\int_{e:c(x,e)=c} \pi_0^{1st}(e \mid x)}.$$

This general version of the POTEC gradient estimator is unbiased under local correctness (i.e, $\Delta_q(x, a, b) = \Delta_{\hat{f}}(x, a, b)$, $\forall x, a, b$ such that $c(x, \Phi(x, a)) = c(x, \Phi(x, b))$). Since the 1st-stage policy is learned in the action representation space, it can naturally exploit the smoothness in $\mathcal{E}$.

If we follow this implementation, in the inference time, for an incoming context $x$, we first sample a point in the action representation space $\mathcal{E}$ from the 1st-stage policy as $e \sim \pi_\theta^{1st}(\cdot \mid x)$, which implies a promising region in $\mathcal{E}$. Note that, in general, $e \in \mathcal{E}$ will not match with any already observed action representation $\{\Phi(x_i, a_i)\}_{i=1}^n$. Then, the second-stage $\pi_\psi^{2nd}$, which is constructed from the pairwise regression model $\hat{h}_\psi : \mathcal{X} \times \mathcal{E} \to \mathbb{R}$, identifies the best action within the promising region as

$$a = \underset{a': c(x, \Phi(x, a')) = c(x, e)}{\arg\max} \hat{h}_\psi(x, \Phi(x, a')),$$

where $\{a' \in \mathcal{A} \mid c(x, \Phi(x, a')) = c(x, e)\}$ is the set of actions whose representation lies in the promising region induced by $e \sim \pi_\theta^{1st}(\cdot \mid x)$.

**Implementation Option 2:** This option first learns a parameterized distribution over the action space $\mathcal{A}$ as the 1st-stage policy using $\Phi(x,a)$ as its input via the following version of the POTEC gradient estimator.

$$\nabla_\theta \widehat{V}_{\text{POTEC}}(\pi_{\theta,\psi}^{overall}; \mathcal{D}) := \frac{1}{n} \sum_{i=1}^n \left\{ w(x_i, c_i)(r_i - \hat{f}(x_i, \Phi(x_i, a_i)))\nabla_\theta \log \pi_\theta(a_i \mid x_i; \Phi(x_i, a_i)) \right.$$
$$\left. + \mathbb{E}_{a \sim \pi_\theta^{1st}}[\hat{f}^{\pi_\psi^{2nd}}(x_i, c)\nabla_\theta \log \pi_\theta(a \mid x_i; \Phi(x_i, a))] \right\},$$
$$(15)$$

where $c_i = c(x_i, \Phi(x_i, a_i))$, $\hat{f}^{\pi_\psi^{2nd}}(x, c) := \mathbb{E}_{\pi_\psi^{2nd}}[\hat{f}(x, a)]$ and

$$w(x, c) := \frac{\pi_\theta^{1st}(c \mid x)}{\pi_0^{1st}(c \mid x)} = \frac{\sum_{a: c(x, \Phi(x, a)) = c} \pi_\theta^{1st}(a \mid x; \Phi(x, a))}{\sum_{a: c(x, \Phi(x, a)) = c} \pi_0^{1st}(a \mid x; \Phi(x, a))}.$$

This version is also unbiased under local correctness (i.e, $\Delta_q(x, a, b) = \Delta_{\hat{f}}(x, a, b)$, $\forall x, a, b$ such that $c(x, \Phi(x, a)) = c(x, \Phi(x, b))$). The 1st-stage policy also simply leverages the action representation as its input.[7]

If we follow this implementation, in the inference time, for an incoming context $x$, we first sample a point in the action space $\mathcal{A}$ from the 1st-stage policy as $a \sim \pi_\theta^{1st}(\cdot \mid x; \Phi(x, a))$, which merely implies a promising region in $\mathcal{E}$. Then, the second-stage $\pi_\psi^{2nd}$, which is constructed from the pairwise regression model $\hat{h}_\psi : \mathcal{X} \times \mathcal{E} \to \mathbb{R}$, identifies the best action within the promising region as

$$a = \underset{a': c(x, \Phi(x, a')) = c(x, \Phi(x, a))}{\arg\max} \hat{h}_\psi(x, \Phi(x, a')),$$

where $\{a' \in \mathcal{A} \mid c(x, \Phi(x, a')) = c(x, \Phi(x, a))\}$ is the set of actions whose representation lies in the promising region induced by $a \sim \pi_\theta^{1st}(\cdot \mid x; \Phi(x, a))$.

The empirical comparison of the above two options highly depends on each application. For example, **Implementation Option 1** may perform better when the action representation space $\mathcal{E}$ is low-dimensional while it may suffer when $\mathcal{E}$ is high-dimensional. Therefore, under the presence of some action representation $\Phi(x, a)$, we would encourage the practitioners to identify the best implementations for their particular application in a data-driven fashion, for example, by performing a careful cross-validation.

## C.1 THE ONE-STAGE VARIANT OF POTEC

It is worth noting that there exists a one-stage variant of POTEC, as opposed to the two-stage variant, which is our primary proposal. More specifically, the one-stage variant directly trains a parameterized overall policy in the action space, $\pi_\theta(a \mid x)$, via the POTEC gradient estimator as follows:

$$\nabla_\theta \widehat{V}_{\text{POTEC1}}(\pi_\theta; \mathcal{D}) := \frac{1}{n} \sum_{i=1}^n \left\{ w(x_i, c_{a_i})(r_i - \hat{f}(x_i, a_i))s_\theta(x_i, a_i) + \mathbb{E}_{\pi_\theta(a \mid x_i)}[\hat{f}(x_i, a)s_\theta(x_i, a)] \right\},$$

where $s_\theta(x, a) := \nabla_\theta \log \pi_\theta(a \mid x)$. Although the one-stage variant is categorized as a policy-based approach, as it trains the overall policy directly via policy gradient, it still achieves significant variance reduction compared to IPS-PG and DR-PG and remains unbiased under local correctness. However, the one-stage variant could be considered a suboptimal utilization of the local correctness condition since, given a locally correct regression model, we should be able to optimally choose the action within a cluster as in Eq. (10) and thus do not need to learn the overall policy solely through policy gradient. Nevertheless, the one-stage variant may be valuable in practice, as it do not need to maintain and execute multiple policies. We provide an empirical comparison of the one-stage and two-stage variants of POTEC in Appendix E.

---

[7]For example, we can define a parameterized policy as

$$\pi_\theta(a \mid x; \Phi(x, a)) = \frac{\exp(f_\theta(x, \Phi(x, a)))}{\sum_{a' \in \mathcal{A}} \exp(f_\theta(x, \Phi(x, a')))}$$

where $f_\theta : \mathcal{X} \times \mathcal{E} \to \mathbb{R}$ is some parameterized function having action representation $\Phi(x, a)$ as its input.

Table 2: Dependence of the cluster value on the 2nd-stage policy ($q^{\pi_\psi^{2nd}}(x,c)$)

| $a$ | $a_0$ | $a_1$ | $a_2$ | $a_3$ |
|---|---|---|---|---|
| $c(x_0, a)$ | 0 | | 1 | |
| $q(x_0, a)$ | 4 | 2 | 5 | 0 |
| $\pi_\psi^{2nd}(a\|x,c)$ | 1 | 0 | 1 | 0 |
| $q^{\pi_\psi^{2nd}}(x,c)$ | 4 | | 5 | |

| $a$ | $a_0$ | $a_1$ | $a_2$ | $a_3$ |
|---|---|---|---|---|
| $c(x_0, a)$ | 0 | | 1 | |
| $q(x_0, a)$ | 4 | 2 | 5 | 0 |
| $\pi_\psi^{2nd}(a\|x,c)$ | 0.5 | 0.5 | 0.5 | 0.5 |
| $q^{\pi_\psi^{2nd}}(x,c)$ | 3 | | 2.5 | |

## D    OMITTED PROOFS

### D.1    DERIVATION OF EQ. (6)

$$\nabla_\theta V\left(\pi_{\theta,\psi}^{overall}\right) = \mathbb{E}_{p(x)}\left[\sum_{a\in\mathcal{A}} q(x,a)\nabla_\theta \pi_{\theta,\psi}^{overall}(a\,|\,x)\right]$$

$$= \mathbb{E}_{p(x)}\left[\sum_{a\in\mathcal{A}} q(x,a)\sum_{c\in\mathcal{C}}\nabla_\theta\pi_\theta^{1st}(c\,|\,x)\pi_\psi^{2nd}(a\,|\,x,c)\right]$$

$$= \mathbb{E}_{p(x)}\left[\sum_{c\in\mathcal{C}}\nabla_\theta\pi_\theta^{1st}(c\,|\,x)\sum_{a\in\mathcal{A}} q(x,a)\pi_\psi^{2nd}(a\,|\,x,c)\right]$$

$$= \mathbb{E}_{p(x)}\left[\sum_{c\in\mathcal{C}}\pi_\theta^{1st}(c\,|\,x)\nabla_\theta\log\pi_\theta^{1st}(c\,|\,x)q^{\pi_\psi^{2nd}}(x,c)\right]$$

$$= \mathbb{E}_{p(x)\pi_\theta^{1st}(c\,|\,x)}\left[q^{\pi_\psi^{2nd}}(x,c)s_\theta(x,c)\right]$$

where we use $q^{\pi_\psi^{2nd}}(x,c) := \mathbb{E}_{\pi_\psi^{2nd}(a|x,c)}[q(x,a)]$ and $s_\theta(x,c) := \nabla_\theta\log\pi_\theta^{1st}(c\,|\,x)$. The above policy gradient suggests increasing the choice probability of a cluster that is promising under the given 2nd-stage policy $\pi_\psi^{2nd}$ where the effectiveness of a cluster under the 2nd-stage policy is quantified by $q^{\pi_\psi^{2nd}}(x,c)$. This implies that the optimal cluster can be different given different 2nd-stage policies. A toy example in Table 2 shows that the value of a cluster can indeed be very different given different 2nd-stage policies. More specifically, the left table shows the case with the optimal 2nd-stage policy that can identify the best action within each cluster. Then, we can see that the optimal cluster is $c = 1$, since the maximum expected reward in the actions of this cluster is larger. In contrast, the right table shows the case with uniform 2nd-stage policy. Under such a 2nd-stage policy, the optimal cluster then becomes $c = 0$, since the average expected reward of the actions in $c = 0$ is larger than that of $c = 1$.

Below we prove the theorems presented in the main text based on the following general version of the POTEC gradient estimator.

$$\nabla_\theta\widehat{V}_{\text{POTEC}}(\pi_{\theta,\psi}^{overall};\mathcal{D}) := \frac{1}{n}\sum_{i=1}^n\left\{w(x_i,c_i)(r_i - \hat{f}(x_i,a_i))s_\theta(x_i,c_a) + \mathbb{E}_{\pi_\theta^{1st}}[\hat{f}^{\pi_\psi^{2nd}}(x_i,c)s_\theta(x_i,c_i)]\right\}$$

where $c_i \sim p(\cdot\,|\,x_i,a_i)$ is a stochastic and context-dependent clustering. The POTEC gradient estimator defined in Eq. (7) can be considered a special case with a deterministic and context-independent clustering function $c : \mathcal{A}\to\mathcal{C}$.

Note that we use $w(x,c) = \mathbb{E}_{\pi(a|x,c)}[w(x,a)]$ and $w(x,a) = \frac{\pi(a\,|\,x)}{\pi_0(a\,|\,x)} = \frac{\pi(a,c\,|\,x)}{\pi_0(a,c\,|\,x)}$ in the following.

### D.2    PROOF OF THEOREM 3.2 AND COROLLARY ??

*Proof.* To derive the bias of the POTEC gradient estimator, we calculate the difference between its expectation and the true policy gradient given in Eq. (6) below.

$$\text{Bias}(\nabla_\theta \widehat{V}_{\text{POTEC}}(\pi_{\theta,\psi}^{overall}; \mathcal{D}))$$

$$= \mathbb{E}_{p(x)\pi_0(a|x)p(c|x,a)p(r|x,a)}[w(x,c)(r - \hat{f}(x,a))s_\theta(x,c)] + \mathbb{E}_{p(x)\pi_\theta^{1st}(c|x)}[\hat{f}^{\pi_\psi^{2nd}}(x,c)s_\theta(x,c)]$$
$$\quad - \mathbb{E}_{p(x)\pi_\theta^{1st}(c|x)}\left[q^{\pi_\psi^{2nd}}(x,c)s_\theta(x,c)\right]$$

$$= \mathbb{E}_{p(x)}\left[\sum_{a\in\mathcal{A}}\pi_0(a\,|\,x)\Delta_{q,\hat{f}}(x,a)\sum_{c\in\mathcal{C}}p(c\,|\,x,a)w(x,c)s_\theta(x,c)\right] + \mathbb{E}_{p(x)}\left[\sum_{c\in\mathcal{C}}\pi_\theta^{1st}(c\,|\,x)\hat{f}^{\pi_\psi^{2nd}}(x,c)s_\theta(x,c)\right]$$
$$\quad - \mathbb{E}_{p(x)}\left[\sum_{c\in\mathcal{C}}\pi_\theta^{1st}(c\,|\,x)q^{\pi_\psi^{2nd}}(x,c)s_\theta(x,c)\right]$$

$$= \mathbb{E}_{p(x)}\left[\sum_{a\in\mathcal{A}}\pi_0(a\,|\,x)\Delta_{q,\hat{f}}(x,a)\sum_{c\in\mathcal{C}}\frac{\pi_0^{1st}(c\,|\,x)\pi_0^{2nd}(a\,|\,x,c)}{\pi_0(a\,|\,x)}w(x,c)s_\theta(x,c)\right]$$
$$\quad + \mathbb{E}_{p(x)}\left[\sum_{c\in\mathcal{C}}\pi_0^{1st}(c\,|\,x)\frac{\pi_\theta^{1st}(c\,|\,x)}{\pi_0^{1st}(c\,|\,x)}\hat{f}^{\pi_\psi^{2nd}}(x,c)s_\theta(x,c)\right] - \mathbb{E}_{p(x)}\left[\sum_{c\in\mathcal{C}}\pi_0^{1st}(c\,|\,x)\frac{\pi_\theta^{1st}(c\,|\,x)}{\pi_0^{1st}(c\,|\,x)}q^{\pi_\psi^{2nd}}(x,c)s_\theta(x,c)\right]$$

$$= \mathbb{E}_{p(x)\pi_0^{1st}(c|x)}\left[w(x,c)s_\theta(x,c)\sum_{a\in\mathcal{A}}\pi_0^{2nd}(a\,|\,x,c)\Delta_{q,\hat{f}}(x,a)\right]$$
$$\quad + \mathbb{E}_{p(x)\pi_0^{1st}(c|x)}\left[w(x,c)s_\theta(x,c)\hat{f}^{\pi_\psi^{2nd}}(x,c)\right] - \mathbb{E}_{p(x)\pi_0^{1st}(c\,|\,x)}\left[w(x,c)s_\theta(x,c)q^{\pi_\psi^{2nd}}(x,c)\right]$$

$$= \mathbb{E}_{p(x)\pi_0^{1st}(c|x)}\left[w(x,c)s_\theta(x,c)\sum_{a\in\mathcal{A}}\pi_0^{2nd}(a\,|\,x,c)\Delta_{q,\hat{f}}(x,a)\right]$$
$$\quad - \mathbb{E}_{p(x)\pi_0^{1st}(c|x)}\left[s_\theta(x,c)\sum_{a\in\mathcal{A}}\frac{\pi_\theta^{1st}(c\,|\,x)}{\pi_0^{1st}(c\,|\,x)}\frac{\pi_\psi^{2nd}(a\,|\,x,c)}{\pi_0^{2nd}(a\,|\,x,c)}\pi_0^{2nd}(a\,|\,x,c)\Delta_{q,\hat{f}}(x,a)\right]$$

$$= \mathbb{E}_{p(x)\pi_0^{1st}(c|x)}\left[s_\theta(x,c)\sum_{a\in\mathcal{A}}w(x,a)\pi_0^{2nd}(a\,|\,x,c)\sum_{b\in\mathcal{A}}\pi_0^{2nd}(b\,|\,x,c)\Delta_{q,\hat{f}}(x,b)\right]$$
$$\quad - \mathbb{E}_{p(x)\pi_0^{1st}(c|x)}\left[s_\theta(x,c)\sum_{a\in\mathcal{A}}w(x,a)\pi_0^{2nd}(a\,|\,x,c)\Delta_{q,\hat{f}}(x,a)\right]$$

$$= \mathbb{E}_{p(x)\pi_0^{1st}(c|x)}\left[s_\theta(x,c)\sum_{a\in\mathcal{A}}w(x,a)\pi_0^{2nd}(a\,|\,x,c)\left(\left(\sum_{b\in\mathcal{A}}\pi_0^{2nd}(b\,|\,x,c)\Delta_{q,\hat{f}}(x,b)\right) - \Delta_{q,\hat{f}}(x,a)\right)\right]$$

where $\Delta_{q,\hat{f}}(x,a) := q(x,a) - \hat{f}(x,a)$. By applying Lemma B.1 of (Saito & Joachims, 2022) to the last line (setting $f(a) = w(,a), g(a) = \pi_0^{2nd}(a\,|\,,), h(a) = \Delta(,a)$), we obtain the following expression of the bias.

$$\mathbb{E}_{p(x)\pi_0^{1st}(c|x)}\left[s_\theta(x,c)\sum_{a<b}\pi_0^{2nd}(a\,|\,x,c)\pi_0^{2nd}(b\,|\,x,c)\left(\Delta_{q,\hat{f}}(x,a) - \Delta_{q,\hat{f}}(x,b)\right)(w(x,b) - w(x,a))\right]$$

In particular, in the simpler case of deterministic and context-independent clustering as in the main text, we can simplify the expression of the bias as below.

$$\mathbb{E}_{p(x)\pi_0^{1st}(c|x)}\left[\sum_{a<b:c_a=c_b=c}\pi_0^{2nd}(a\,|\,x,c)\pi_0^{2nd}(b\,|\,x,c)\left(\Delta_{q,\hat{f}}(x,a) - \Delta_{q,\hat{f}}(x,b)\right)(w(x,b) - w(x,a))\,s_\theta(x,c)\right]$$

$$= \mathbb{E}_{p(x)\pi_0^{1st}(c|x)}\left[\sum_{a<b:c_a=c_b=c}\pi_0^{2nd}(a\,|\,x,c)\pi_0^{2nd}(b\,|\,x,c)\left(\Delta_q(x,a,b) - \Delta_{\hat{f}}(x,a,b)\right)(w(x,b) - w(x,a))\,s_\theta(x,c)\right]$$

where, we used $\pi_0^{2nd}(a\,|\,x,c) = \frac{\pi_0(a\,|\,x)\mathbb{I}\{c_a=c\}}{\pi_0^{1st}(c\,|\,x)}$ and $\Delta_{q,\hat{f}}(x,a) - \Delta_{q,\hat{f}}(x,b) \Rightarrow \Delta_q(x,a,b) - \Delta_{\hat{f}}(x,a,b)$. $\qquad\square$

### D.3 PROOF OF PROPOSITION 3.4

*Proof.* We apply the law of total variance several times to obtain the variance of the $j$-th element of the POTEC gradient estimator for a particular parameter $\theta \in \mathbb{R}^d$ in the following.

$$\mathbb{V}_{p(x)\pi_0(a|x)p(c|x,a)p(r|x,a)} \left[ w(x,c)(r - \hat{f}(x,a))s_\theta^{(j)}(x,c) + \mathbb{E}_{\pi_\theta^{1st}(c'|x)}[\hat{f}^{\pi_\psi^{2nd}}(x,c')s_\theta^{(j)}(x,c')] \right]$$

$$= \mathbb{E}_{p(x)\pi_0(a|x)p(c|x,a)} \left[ \mathbb{V}_{p(r|x,a)} \left[ w(x,c)(r - \hat{f}(x,a))s_\theta^{(j)}(x,c) + \mathbb{E}_{\pi_\theta^{1st}(c'|x)}[\hat{f}^{\pi_\psi^{2nd}}(x,c')s_\theta^{(j)}(x,c')] \right] \right]$$

$$+ \mathbb{V}_{p(x)\pi_0(a|x)p(c|x,a)} \left[ \mathbb{E}_{p(r|x,a)} \left[ w(x,c)(r - \hat{f}(x,a))s_\theta^{(j)}(x,c) + \mathbb{E}_{\pi_\theta^{1st}(c'|x)}[\hat{f}^{\pi_\psi^{2nd}}(x,c')s_\theta^{(j)}(x,c')] \right] \right]$$

$$= \mathbb{E}_{p(x)\pi_0(a|x)p(c|x,a)} \left[ (w(x,c)s_\theta^{(j)}(x,c))^2 \sigma^2(x,a) \right]$$

$$+ \mathbb{V}_{p(x)\pi_0(a|x)p(c|x,a)} \left[ w(x,c)\Delta_{q,\hat{f}}(x,a)s_\theta^{(j)}(x,c) + \mathbb{E}_{\pi_\theta^{1st}(c'|x)}[\hat{f}^{\pi_\psi^{2nd}}(x,c')s_\theta^{(j)}(x,c')] \right]$$

$$= \mathbb{E}_{p(x)\pi_0(a|x)p(c|x,a)} \left[ (w(x,c)s_\theta^{(j)}(x,c))^2 \sigma^2(x,a) \right]$$

$$+ \mathbb{E}_{p(x)\pi_0(a|x)} \left[ \mathbb{V}_{p(c|x,a)} \left[ w(x,c)\Delta_{q,\hat{f}}(x,a)s_\theta^{(j)}(x,c) + \mathbb{E}_{\pi_\theta^{1st}(c'|x)}[\hat{f}^{\pi_\psi^{2nd}}(x,c')s_\theta^{(j)}(x,c')] \right] \right]$$

$$+ \mathbb{V}_{p(x)\pi_0(a|x)} \left[ \mathbb{E}_{p(c|x,a)} \left[ w(x,c)\Delta_{q,\hat{f}}(x,a)s_\theta^{(j)}(x,c) + \mathbb{E}_{\pi_\theta^{1st}(c'|x)}[\hat{f}^{\pi_\psi^{2nd}}(x,c')s_\theta^{(j)}(x,c')] \right] \right]$$

$$= \mathbb{E}_{p(x)\pi_0(a|x)p(c|x,a)} \left[ (w(x,c)s_\theta^{(j)}(x,c))^2 \sigma^2(x,a) \right] + \mathbb{E}_{p(x)\pi_0(a|x)} \left[ \mathbb{V}_{p(c|x,a)} \left[ w(x,c)\Delta_{q,\hat{f}}(x,a)s_\theta^{(j)}(x,c) \right] \right]$$

$$+ \mathbb{V}_{p(x)\pi_0(a|x)} \left[ \mathbb{E}_{p(c|x,a)} \left[ w(x,c)\Delta_{q,\hat{f}}(x,a)s_\theta^{(j)}(x,c) \right] + \mathbb{E}_{\pi_\theta^{1st}(c'|x)}[\hat{f}^{\pi_\psi^{2nd}}(x,c)s_\theta^{(j)}(x,c')] \right]$$

$$= \mathbb{E}_{p(x)\pi_0(a|x)p(c|x,a)} \left[ (w(x,c)s_\theta^{(j)}(x,c))^2 \sigma^2(x,a) \right] + \mathbb{E}_{p(x)\pi_0(a|x)} \left[ \mathbb{V}_{p(c|x,a)} \left[ w(x,c)\Delta_{q,\hat{f}}(x,a)s_\theta^{(j)}(x,c) \right] \right]$$

$$+ \mathbb{E}_{p(x)} \left[ \mathbb{V}_{\pi_0(a|x)} \left[ \mathbb{E}_{p(c|x,a)} \left[ w(x,c)\Delta_{q,\hat{f}}(x,a)s_\theta^{(j)}(x,c) \right] + \mathbb{E}_{\pi_\theta^{1st}(c'|x)}[\hat{f}^{\pi_\psi^{2nd}}(x,c')s_\theta^{(j)}(x,c')] \right] \right]$$

$$+ \mathbb{V}_{p(x)} \left[ \mathbb{E}_{\pi_0(a|x)} \left[ \mathbb{E}_{p(c|x,a)} \left[ w(x,c)\Delta_{q,\hat{f}}(x,a)s_\theta^{(j)}(x,c) \right] + \mathbb{E}_{\pi_\theta^{1st}(c'|x)}[\hat{f}^{\pi_\psi^{2nd}}(x,c')s_\theta^{(j)}(x,c')] \right] \right]$$

$$= \mathbb{E}_{p(x)\pi_0(a|x)p(c|x,a)} \left[ (w(x,c)s_\theta^{(j)}(x,c))^2 \sigma^2(x,a) \right] + \mathbb{E}_{p(x)\pi_0(a|x)} \left[ \mathbb{V}_{p(c|x,a)} \left[ w(x,c)\Delta_{q,\hat{f}}(x,a)s_\theta^{(j)}(x,c) \right] \right]$$

$$+ \mathbb{E}_{p(x)} \left[ \mathbb{V}_{\pi_0(a|x)} \left[ \mathbb{E}_{p(c|x,a)} \left[ w(x,c)\Delta_{q,\hat{f}}(x,a)s_\theta^{(j)}(x,c) \right] \right] \right] + \mathbb{V}_{p(x)} \left[ \mathbb{E}_{\pi_\theta^{1st}(c|x)} \left[ q^{\pi_\psi^{2nd}}(x,c)s_\theta^{(j)}(x,c) \right] \right],$$

where we rely on local correctness in the last line to use

$$\mathbb{E}_{\pi_0(a|x)} \left[ \mathbb{E}_{p(c|x,a)} \left[ w(x,c)\Delta_{q,\hat{f}}(x,a)s_\theta^{(j)}(x,c) \right] + \mathbb{E}_{\pi_\theta^{1st}(c'|x)}[\hat{f}^{\pi_\psi^{2nd}}(x,c')s_\theta^{(j)}(x,c')] \right] = \mathbb{E}_{\pi_\theta^{1st}(c|x)} \left[ q^{\pi_\psi^{2nd}}(x,c)s_\theta^{(j)}(x,c) \right].$$

In particular, in the case of deterministic and context-independent clustering, the variance can be simplified as follows.

$$\mathbb{V}_{p(x)\pi_0(a|x)p(r|x,a)} \left[ w(x,c_a)(r - \hat{f}(x,a))s_\theta^{(j)}(x,c_a) + \mathbb{E}_{\pi_\theta^{1st}(c'|x)}[\hat{f}^{\pi_\psi^{2nd}}(x,c)s_\theta^{(j)}(x,c')] \right]$$

$$= \mathbb{E}_{p(x)\pi_0(a|x)} \left[ (w(x,c_a)s_\theta^{(j)}(x,c_a))^2 \sigma^2(x,a) \right]$$

$$+ \mathbb{E}_{p(x)} \left[ \mathbb{V}_{\pi_0(a|x)} \left[ w(x,c_a)\Delta_{q,\hat{f}}(x,a)s_\theta^{(j)}(x,c_a) \right] \right] + \mathbb{V}_{p(x)} \left[ \mathbb{E}_{\pi_\theta^{1st}(c|x)} \left[ q^{\pi_\psi^{2nd}}(x,c)s_\theta^{(j)}(x,c) \right] \right].$$

$$\square$$

### D.4 PROOF OF PROPOSITION 3.5

*Proof.* Since $\mathbb{E}_{p(x)\pi_0(a|x)p(c|x,a)}[w(x,c)] = \mathbb{E}_{p(x)\pi_0(a|x)p(c|x,a)}[w(x,a)] = 1$, the difference in the variance of the cluster and vanilla importance weights is attributed to the difference in their second

Table 3: Hyperparameter search spaces used in the experiments. $\lambda$ is the hyperparameter for weight decay. $\eta$ is the learning rate. $B$ is the batch size.

| Datasets | Methods | $\lambda$ | $\eta$ | $B$ | $|\Phi(x)|$ in Eq.(12) |
|---|---|---|---|---|---|
| Synthetic | IPS-PG | $\{10^{-2}, 10^{-4}, 10^{-6}\}$ | $\{10^{-3}, 5 \times 10^{-4}, 10^{-4}\}$ | $\{64, 128, 256\}$ | $\{0.1|\mathcal{A}|, 0.5|\mathcal{A}|, |\mathcal{A}|\}$ |
| | DR-PG | $\{10^{-2}, 10^{-4}, 10^{-6}\}$ | $\{10^{-3}, 5 \times 10^{-4}, 10^{-4}\}$ | $\{64, 128, 256\}$ | $\{0.1|\mathcal{A}|, , 0.5|\mathcal{A}|, |\mathcal{A}|\}$ |
| | POCEM | $10^{-4}$ | $5 \times 10^{-4}$ | $128$ | - |
| Real-World | IPS-PG | $[10^{-4}, 10^{-2}]$ | $[10^{-4}, 10^{-2}]$ | $1,024$ | $|\mathcal{A}|$ |
| | DR-PG | $[10^{-4}, 10^{-2}]$ | $[10^{-4}, 10^{-2}]$ | $1,024$ | $|\mathcal{A}|$ |
| | POCEM | $10^{-4}$ | $10^{-3}$ | $1,024$ | - |

moment, which is calculated below.

$$\mathbb{V}_{p(x)\pi_0(a|x)p(c|x,a)}[w(x,a)] - \mathbb{V}_{p(x)\pi_0(a|x)p(c|x,a)}[w(x,c)]$$

$$= \mathbb{E}_{p(x)\pi_0(a|x)p(c|x,a)}[w^2(x,a)] - \mathbb{E}_{p(x)\pi_0(a|x)p(c|x,a)}[w^2(x,c)]$$

$$= \mathbb{E}_{p(x)\pi_0(a|x)p(c|x,a)}\left[w^2(x,a) - \left(\mathbb{E}_{\pi_0(a|x,c)}[w(x,a)]\right)^2\right] \quad \because w(x,c) = \mathbb{E}_{\pi_0(a|x,c)}[w(x,a)]$$

$$= \mathbb{E}_{p(x)}\left[\sum_{a \in \mathcal{A}} \pi_0(a|x) \sum_{c \in \mathcal{C}} p(c|x,a) \left(w^2(x,a) - \left(\mathbb{E}_{\pi_0(a|x,c)}[w(x,a)]\right)^2\right)\right]$$

$$= \mathbb{E}_{p(x)}\left[\sum_{a \in \mathcal{A}} \pi_0(a|x) \sum_{c \in \mathcal{C}} \frac{\pi_0(c|x)\pi_0(a|x,c)}{\pi_0(a|x)} \left(w^2(x,a) - \left(\mathbb{E}_{\pi_0(a|x,c)}[w(x,a)]\right)^2\right)\right] \quad \because p(c|x,a) = \frac{\pi_0(c|x)\pi_0(a|x,c)}{\pi_0(a|x)}$$

$$= \mathbb{E}_{p(x)\pi_0(c|x)}\left[\mathbb{E}_{\pi_0(a|x,c)}[w^2(x,a)] - \left(\mathbb{E}_{\pi_0(a|x,c)}[w(x,a)]\right)^2\right]$$

$$= \mathbb{E}_{p(x)\pi_0(c|x)}\left[\mathbb{V}_{\pi_0(a|x,c)}[w(x,a)]\right]$$

$\square$

# E    ADDITIONAL EXPERIMENT SETUPS AND RESULTS

## E.1    SYNTHETIC EXPERIMENT

**Detailed Setup.**    This section describes how we define the synthetic reward function and perform hyperparameter tuning in detail. Recall that, in the synthetic experiment, we synthesized the expected reward function as

$$q(x,a) = g(x, c_a) + h_{c_a}(x,a), \tag{16}$$

where we use the following functions as $g$ (cluster effect) and $h$ (residual effect), respectively.

$$g(x, c_a) = g_{base}(x, c_a) + u_1 \mathbb{I}\{(\sum_{d=1}^{3} x_d) < 1.5\}$$

$$+ u_2 \mathbb{I}\{(\sum_{d=3}^{8} x_d) < -0.5\} + u_3 \mathbb{I}\{(\sum_{d=2}^{3} x_d) > 3.0\} + u_4 \mathbb{I}\{(\sum_{d=5}^{10} x_d) < 1.0\},$$

$$h_{c_a}(x,a) = x^\top M_{c_a} \text{one\_hot}_a + \theta_{x,c_a}^\top x + \theta_{a,c_a}^\top \text{one\_hot}_a,$$

where $x_d$ is the $d$-th dimension of the context vector $x$. We use **obp.dataset.polynomial_reward_function** from OpenBanditPipeline[8] as $g_{base}(\cdot, \cdot)$ and $u_1, \ldots, u_4$ are sampled from a uniform distribution with range $[-3, 3]$. $M_{c_a}$, $\theta_{x,c_a}$, and $\theta_{a,c_a}$ are parameter matrices or vectors sampled from a uniform distribution with range $[-1, 1]$ separately for each given action cluster $c_a$.

Based on the above reward function $q(x,a)$, we synthesized the logging policy $\pi_0$ as

$$\pi_0(a \mid x) = \frac{\exp(\beta \cdot q(x,a) + \eta_{x,a})}{\sum_{a' \in \mathcal{A}} \exp(\beta \cdot q(x,a') + \eta_{x,a})}, \tag{17}$$

---

[8]https://github.com/st-tech/zr-obp

Table 4: Dataset Statistics

| **Dataset** | $n_{train}$ | $n_{test}$ | $|\mathcal{A}|$ |
|---|---|---|---|
| EUR-Lex 4K | 15,449 | 3,865 | 3,956 |
| Wiki10-31K | 14,146 | 6,616 | 30,938 |

where $\beta$ is a parameter that controls the optimality of the logging policy. $\beta$ is set to 5 as default and the noise $\eta_{x,a}$ is sampled from a normal distribution.

To summarize, we first sample a context and define the expected reward $q(x, a)$ as in Eq. (16). We then sample discrete action $a$ from $\pi_0$ based on Eq. (17) where action $a$ is associated with a cluster $c_a$. The reward is then sampled from a normal distribution with mean $q(x, a)$. Iterating this procedure $n$ times generates logged data $\mathcal{D}$ with $n$ independent copies of $(x, a, c_a, r)$.

We tuned the weight decay hyperparameter, learning rate, batch size, and the number of irrelevant actions for variance reduction for the baseline methods (i.e., IPS-PG and DR-PG) using the test policy value, while we use a fixed set of hyperparameters for POTEC as shown in Table 3, giving an unfair advantage to the baselines. For all methods, we used Adam (Kingma & Ba, 2014) as the optimizer and used neural networks with 3 hidden layers to parameterize the policy.

## E.2 EXPERIMENT ON EXTREME CLASSIFICATION DATA

In addition to synthetic and real-world recommendation data, we performed OPL experiments on two extreme classification datasets provided by Bhatia et al. (2016).

**Setup.** Following previous studies (Dudík et al., 2014; Saito et al., 2021b; Su et al., 2020a; Wang et al., 2017), we transform the extreme classification datasets to contextual bandit feedback data with many actions. In a classification dataset $\{(x_i, a_i)\}_{i=1}^n$, we have some feature vector $x_i \in \mathcal{X}$ and ground-truth label $a_i \in \mathcal{A}$, which will be considered an action.

We consider stochastic continuous rewards where we define the expected reward function as follows.

$$q(x, a) = \begin{cases} 1 - \eta_a & \text{if } a \text{ is a positive label} \\ \eta_a & \text{otherwise} \end{cases} \tag{18}$$

$\eta_a$ is a noise parameter sampled separately for each action $a$ from a uniform distribution with range $[0, 0.1]$. After defining the expected reward function, we sample the reward from a normal distribution as $r \sim \mathcal{N}(q(x, a), \sigma^2)$ with standard deviation $\sigma = 0.05$ for each data.

We define the logging policy $\pi_0$ by applying softmax to an estimated reward function $\tilde{q}(x, a)$ as

$$\pi_0(a \mid x) = \frac{\exp(\beta \cdot \tilde{q}(x, a))}{\sum_{a' \in \mathcal{A}} \exp(\beta \cdot \tilde{q}(x, a'))}, \tag{19}$$

where we use $\beta = 10$ for both datasets. We obtain $\tilde{q}(x, a)$ by learning a matrix factorization model where we use the test data recorded in the original datasets for obtaining a logging policy while we use the training data for performing OPL to make them independent.

**Results.** We evaluate POTEC against IPS-PG, DR-PG, and Reg-based under varying numbers of clusters to evaluate POTEC's robustness to the choice of this key hyper-parameter. We optimize the hyperparameters of POTEC and the baselines based on the ground-truth policy value in the validation set, and the effectiveness of the OPL methods is evaluated on the test set. For POTEC, we evaluate it with two types of clustering methods to investigate its robustness to the ways the clustering is performed. The first method is through learning an action embedding via Lipschitz regularization (Lip) recently proposed for improving OPE in large action spaces (Peng et al., 2023). The second method is to apply Agglomerative clustering (AC) implemented in scikit-learn (Pedregosa et al., 2011) to the full-information labels, which provides an even more accurate clustering by leveraging the true reward correlation. Note that we perform a conventional reward regression rather than the two-step regression for POTEC due to insufficient pairwise data in these specific datasets.

Figures 6 and 7 report the test policy value (normalized by $V(\pi_0)$) of the OPL methods with varying numbers of clusters on Eurlex-4K and Wiki10-31K, using two types of logging policies. For these

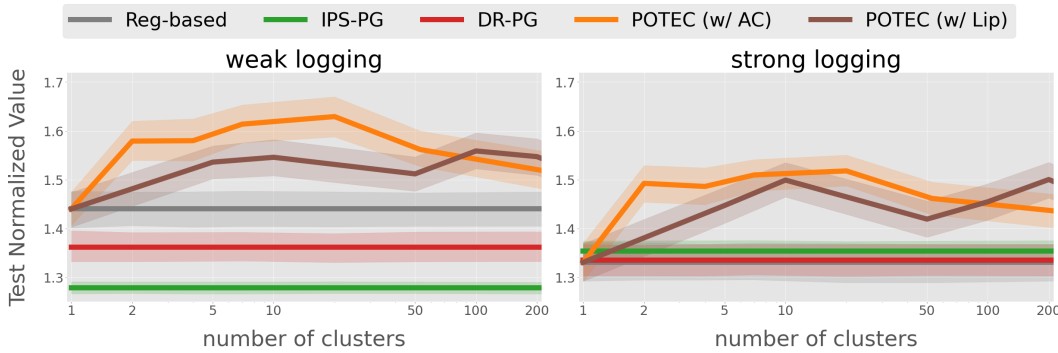

Figure 6: Comparing the test policy value of the OPL methods (normalized by $V(\pi_0)$) on the Eurlex-4K dataset with weak and strong logging policies, respectively.

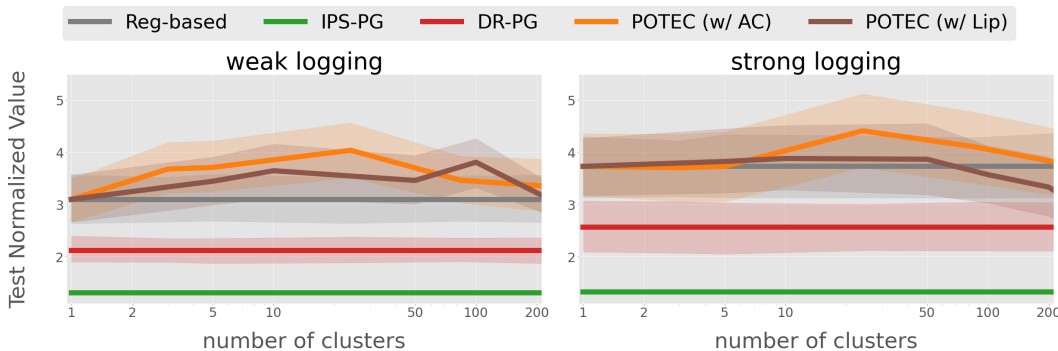

Figure 7: Comparing the test policy value of the OPL methods (normalized by $V(\pi_0)$) on the Wiki10-31K dataset with weak and strong logging policies, respectively.

*Note*: The results are averaged over 5 different sets of synthetic logged data replicated with different random seeds. The shaded regions in the plots represent the 95% confidence intervals of the policy value estimated with bootstrap.

experiments, we trained a "weak logging" policy with (two times) fewer samples than the "strong logging" policy. We optimized the hyperparameters of POTEC and the baselines based on the ground-truth policy value in the validation set, and the effectiveness of the OPL methods is evaluated on the test set. It should be noted that the baseline methods do not depend on action clusters, which results in flat lines in the figures.

The figures demonstrate that POTEC, with both clustering methods (Lipschitz regularization; Lip and Agglomerative clustering; AC), typically outperforms all baseline methods across a range of numbers of clusters, indicating its potential for real-world applications even with action clustering learned only from observable logged data (i.e., POTEC w/ Lip). We can also see that POTEC with a more accurate clustering (i.e., POTEC w/ AC) slightly outperforms POTEC w/ Lip, implying an even better potential of POTEC with a more refined clustering procedure. The regression-based method performs competitively with POTEC only for a strong logging policy on the Wiki10-31K dataset, but we can see, in all other scenarios, POTEC typically performs the best. We also compared the one-stage and two-stage variants of POTEC on the real-world datasets, but we did not find a significant difference between them for both types of clustering.

