# OpenReview forum: "POTEC: Off-Policy Contextual Bandits for Large Action Spaces via Policy Decomposition"
_ICLR.cc/2025/Conference — ICLR 2025 Spotlight_

### Official Review · Reviewer_p75W · 2024-11-01

**Soundness:** 3
**Presentation:** 3
**Contribution:** 3
**Rating:** 8
**Confidence:** 3

**Summary:**

This paper provides a solution to off-policy learning for large action spaces through a two-stage policy. The action space is first clustered with action embedding (reward). Then, the first-stage policy selects an action cluster, and the second-stage policy selects an action within the cluster. The two-stage policy is updated with the policy gradient method, providing a bias and variance analysis of the gradient. They show that the bias of the gradient is controlled by the relative difference in the reward estimation and can be further reduced with a pairwise regression if possible. Also, they show that the variance of the gradient is significantly smaller than the gradient of the original action space (which is straightforward). Empirical studies demonstrate its effectiveness.

**Strengths:**

1, propose a novel new two-stage policy for OPL for bandits with large action space and the intuition/motivation is well-justified.

2, provide a variance and bias analysis of the gradient of the proposed algorithm and connect with the algorithm. Also, the proposed gradient has a smaller variance than the IPS method.

3, the paper is well-written, solid, easy to follow.

**Weaknesses:**

1, The paper did not provide very detailed description of the clustering algorithm. More details will be preferred.

2, The paper did not discuss how the number of the cluster 'C' can affect the performance of the algorithm.

3, They did not experiment with other logging policy settings.

4, Others, see questions.

**Questions:**

1, How do you choose the number of cluster C?

2, how do you get the embedding of action a? Is it from the reward estimation? (more details of the clustering is beneficial)

3, do you assume the logging policy $\pi_0$ known? What if it is not known?

4, Is the logging dataset sampled based on the ground-truth cluster? If I understand correctly when creating the logging dataset, you first cluster the action based on the reward of $(x,a_i)$, and then you have a reward for both cluster r(x,c_a) and r(x,a|c). What if the reward is not in this hierarchy way? For example, if the reward function is just defined as r(x,a) instead of the reward of cluster + reward of the action. Or, in other words, what if the logging policy is not defined as Eq.(11)? I believe that this algorithm might benefit from the logging policy and reward definition while it might not work that well in a more general logging policy. Because the logging policy itself is somehow a two-stage policy.

I am willing to further raise my score if these questions are addressed.

---

> ### Author Response · Authors · 2024-11-16
>
> We appreciate the valuable and thoughtful feedback from the reviewer. We respond to the concrete questions and comments in detail below.
>
> > The paper did not discuss how the number of the cluster 'C' can affect the performance of the algorithm.
>
> We have indeed discussed the effect of the cluster size $C$ from both methodological and empirical perspectives.
>
> Firstly, in Section 3.3 on pages 7–8, we provide an interesting interpretation of our POTEC algorithm as a generalization of the policy-based and regression-based approaches. Here, the cluster size $C$ controls the extent to which POTEC relies on policy-based or regression-based components. Specifically, when $C$ is large, POTEC relies more on its policy-based component (i.e., the first stage). In contrast, using a small $C$ means that we rely more on the regression-based component (i.e., the second stage).
>
> Secondly, we empirically evaluate the effect of $C$ on the performance of POTEC on Figure 5(ii). The figure suggests that POTEC is most sample-efficient with a moderate number of clusters, such as $|C| = 50$ or $100$. It is also noteworthy that POTEC does not underperform the baseline or logging policies even in the worst cases, such as when using overly small ($|C| = 10$) or large ($|C| = 1,000$) cluster spaces.
>
>
> We are more than happy to discuss this point further if our response does not fully address the reviewer’s question.
>
> > 1. How do you choose the number of cluster C?
>
> This is indeed an important point to clarify. The number of clusters is a hyperparameter of POTEC, so we can tune it using the standard cross-validation procedure. However, it is important to perform careful off-policy evaluation (OPE) on validation data to evaluate the performance of each hyperparameter with low bias. To address the variance issue in large action spaces during validation, we recommend using OPE methods from:
>
>
> - Noveen Sachdeva, Lequn Wang, Dawen Liang, Nathan Kallus, and Julian McAuley. Off-policy evaluation for large action spaces via policy convolution. WWW2024
> - Jie Peng, Hao Zou, Jiashuo Liu, Shaoming Li, Yibao Jiang, Jian Pei, and Peng Cui. Offline policy evaluation in large action spaces via outcome-oriented action grouping. WWW2023.
> - Yuta Saito, Ren Qingyang, and Thorsten Joachims. Off-policy evaluation for large action spaces via conjunct effect modeling. ICML2023.
>
>
>
> These papers provide OPE estimators to deal with the variance issue in large action spaces. We will clarify how we can perform hyperparameter tuning for POTEC in the revised version of the paper.
>
> > 2, how do you get the embedding of action a? Is it from the reward estimation? (more details of the clustering is beneficial)
>
> Yes, in our experiments, we used the averaged estimated rewards for each action, $\hat{q}(a):=(1 / n) \sum_{i=1}^{n} \hat{q}\left(x_{i}, a\right)$, as the embedding of $a$ when performing action clustering.
>
> This heuristic method for action clustering worked well and was sufficient to outperform existing OPE methods across a range of scenarios in both synthetic and real-world experiments. However, we believe a more principled and effective approach to action clustering for our method likely exists. As mentioned in Section 5, we consider the development of a clustering algorithm tailored to OPL as valuable future work.
>
> > 3, do you assume the logging policy known? What if it is not known?
>
> Thank you for the great question. We do not necessarily assume that the logging policy is known. In real-world applications, we encounter both known and unknown logging policies. In the relevant literature, it is standard to estimate the logging policy by predicting the action $a$ from the context $x$ on the logged data, when the logging policy is unknown. Similarly, we can estimate the cluster distribution under the logging policy $\pi_0(c|x)$ used in our method by predicting $c$ from $x$ on the logged data.
>
> It is worth noting that, when we need to estimate the logging policy for our proposed method, we must also estimate it to implement existing methods such as IPS-PG and DR-PG, producing some bias in the gradient estimate. We will clarify this in the revision.

---

> > ### Author Response · Authors · 2024-11-16
> > **Official Comment by Authors (Cont'd)**
> >
> > > 4, Is the logging dataset sampled based on the ground-truth cluster? If I understand correctly when creating the logging dataset, you first cluster the action based on the reward of …
> >
> >
> > Thank you for raising this interesting point; it is indeed worth clarifying.
> >
> > First, it is important to emphasize that our algorithm is agnostic to the construction of the underlying reward function. In the synthetic experiment, we defined the reward function by specifying the $g$ (cluster effect) and $h$ (residual effect) functions separately, but our method is applicable to any other underlying reward function. In fact, POTEC is feasible and performs effectively on the KuaiRec dataset in our real-world experiment, where we do not know the functional form of $q(x,a)$.
> >
> >
> > Regarding the logging policy, it is important to note that Eq. (11) is not two-stage, as it directly defines the conditional action distribution rather than the cluster distribution. We can still perform cluster importance weighting with such a logging policy by following the equation $\pi_0(c|x) = \sum_a \mathbb{I} \\{c_a = c \\}\pi_0(a|x)$ to calculate the denominator of the cluster importance weight. However, we agree that the reviewer’s suggestion to test the robustness of our method with other types of logging policies would be an interesting ablation study.

---

> ### Comment · Reviewer_p75W · 2024-11-18
>
> Thanks for the clarification. I raise my score to 8. And it would be beneficial to include more experiments about other logging policies.

---

> > ### Author Response · Authors · 2024-11-27
> >
> > We would like to thank the reviewer for their response and for carefully evaluating our contributions. We agree that an additional ablation study of testing the robustness of our method against varying logging policies would be interesting, and we can implement such an experiment based on our current implementation. If the reviewer has any remaining questions or comments, we would be more than happy to discuss them.

---

### Official Review · Reviewer_5MGC · 2024-11-03

**Soundness:** 3
**Presentation:** 3
**Contribution:** 2
**Rating:** 6
**Confidence:** 3

**Summary:**

This work studies the problem of off-policy learning for contextual bandit policies in large discrete action spaces. The authors propose an approach that decomposes the overall policy into two components: a cluster-selection policy and a conditional action-selection policy. To optimize the cluster-selection policy, they derive a policy gradient, while a pairwise regression procedure is used to learn the conditional action-selection policy. The authors provide a thorough analysis of the bias and variance of the proposed policy gradient estimator, which makes the work more solid and convincing. The authors perform comprehensive numerical studies to demonstrate the performance of the proposed algorithm compared to several baselines.

**Strengths:**

1. The authors study an important problem of off-policy learning for contextual bandit policies in large discrete action spaces.
2. The theoretical analysis of the bias and variance for policy gradient estimator makes the work more solid and convincing.
3. The authors perform comprehensive numerical studies to demonstrate the performance of the proposed algorithm

**Weaknesses:**

The clarity of the writing could be further improved by providing more details on certain aspects, such as practicability of the algorithm and experiment settings. See Questions section.

**Questions:**

1.	The selection of the number of clusters is a crucial aspect of the proposed method, as illustrated in the numerical studies. The author could provide more details on how to select the number of clusters in the real-world applications.
2.	Please provide more details on how to incorporate context-dependent clustering into the algorithm.
3.	The authors could compare the proposed method with Saito et al.,2023, highlighting performance differences and potential advantages/disadvantages.
4.	Please provide more details on data generating mechanism in the real-world experiment.
5.	The authors could consider mentioning potential limitations of the proposed method.

---

> ### Author Response · Authors · 2024-11-16
>
> We appreciate the valuable and thoughtful feedback from the reviewer. We respond to the concrete questions and comments in detail below.
>
> >  The selection of the number of clusters is a crucial aspect of the proposed method, as illustrated in the numerical studies. The author could provide more details on how to select the number of clusters in the real-world applications.
>
>
>
> This is indeed an important point to clarify. The number of clusters is a hyperparameter of POTEC, so we can tune it using the standard cross-validation procedure. However, it is important to perform careful off-policy evaluation (OPE) on validation data to evaluate the performance of each hyperparameter with low bias. To address the variance issue in large action spaces during validation, we recommend using OPE methods from:
>
>
> - Noveen Sachdeva, Lequn Wang, Dawen Liang, Nathan Kallus, and Julian McAuley. Off-policy evaluation for large action spaces via policy convolution. WWW2024
> - Jie Peng, Hao Zou, Jiashuo Liu, Shaoming Li, Yibao Jiang, Jian Pei, and Peng Cui. Offline policy evaluation in large action spaces via outcome-oriented action grouping. WWW2023.
> - Yuta Saito, Ren Qingyang, and Thorsten Joachims. Off-policy evaluation for large action spaces via conjunct effect modeling. ICML2023.
>
>
>
> These papers provide OPE estimators to deal with the variance issue in large action spaces. We will clarify how we can perform hyperparameter tuning for POTEC in the revised version of the paper.
>
> >  Please provide more details on how to incorporate context-dependent clustering into the algorithm.
>
>
>
> This is a good point. It is straightforward to incorporate context-dependent clustering into our algorithm. This can be achieved by applying a context-dependent clustering function, denoted as $c_{x,a}$, instead of the context-independent clustering $c_a$ in Eq. (7) and Eq. (10) when performing first- and second-stage policy learning for POTEC. We will clarify this in the revision.
>
>
> >  The authors could compare the proposed method with Saito et al.,2023, highlighting performance differences and potential advantages/disadvantages.
>
>
>
> We would like to clarify that Saito et al. (2023) focuses solely on the problem of off-policy **evaluation**, not off-policy **learning** as in our work, as described in Section 1 on page 2. Therefore, **our method and that of Saito et al. (2023) are NOT comparable**.
>
> >  Please provide more details on data generating mechanism in the real-world experiment.
>
>
> It is worth noting that the Kuairec dataset used in our real-world experiment is an actual real-world dataset, not a synthetically generated one. The dataset collection process is described in detail in the following paper, which we have already cited:
>
> - Chongming Gao, Shijun Li, Wenqiang Lei, Jiawei Chen, Biao Li, Peng Jiang, Xiangnan He, Jiaxin Mao, and Tat-Seng Chua. Kuairec: A fully-observed dataset and insights for evaluating recommender systems. CIKM 2022.
>
>
> Additionally, the dataset schema is thoroughly described on their website: https://kuairec.com/
>
>
> Using the dataset, we first randomly split the users into training (50%) and test (50%) groups. Then, we draw $n$ users ($x_i$) from the training set with replacement. For each context, we sample action $a_i$ from the logging policy and then sample the reward $r_i$ given $(x_i, a_i)$. This constructs the training logged dataset of the form $\mathcal{D} = \\{(x_i, a_i, r_i)\\}_{i=1}^n$. After training the policies on the train data, we evaluate their performance on the test set using the true reward function $q(x, a)$. This process is repeated 30 times with different draws in the training group.
>
>
> We are more than happy to clarify our real-world experiment procedure further if the reviewer has additional clarification questions.
>
> >  The authors could consider mentioning potential limitations of the proposed method.
>
>
> Thank you for the valuable suggestion. There are indeed limitations to the proposed method, which we could not fully discuss in the main text due to space constraints. For example, using the averaged estimated rewards for each action to perform clustering was sufficient to outperform existing OPL methods, but is a heuristic approach. Thus, there remains room for improvement in the clustering method, as implied by the empirical comparison between POTEC w/ true clusters and w/ learned clusters performed in the synthetic experiment. We will clarify this point further in the revision.

---

> > ### Comment · Reviewer_5MGC · 2024-11-27
> >
> > I would like to thank the author(s) for the responses. I maintain my rating of 6（there is no option 7). I suggest expanding the section on context-dependent clustering with a practical example using either simulated or real-world data. This addition would help readers better understand and implement these methods in their own work.

---

> > > ### Author Response · Authors · 2024-11-27
> > >
> > > We would like to thank the reviewer for their response and for the very useful suggestion. To clarify, we mentioned context-dependent clustering to demonstrate the generality of our framework, i.e., it can be combined with any action clustering method, whether context-independent or context-dependent. Therefore, we did not necessarily propose to use context-dependent clustering in our work.
> > >
> > > However, we agree that it would be helpful to provide an example of context-dependent clustering for clarity. For instance, context-dependent clustering ($c_{x,a}$) can be performed by applying an off-the-shelf algorithm to $\hat{q}(x,a)$ instead of $\hat{q}(a)$, to maximize or minimize the intra-cluster variance of $\hat{q}(x,a)$ for each $x$. We can use the standard cross-validation procedure to identify the best clustering strategy in practice.
> > >
> > > We have demonstrated that POTEC outperforms baseline methods across a range of experimental setups, even with a simple clustering approach, which is context-independent. We believe that developing a more principled (possibly context-dependent) clustering method tailored to OPL is a very interesting direction for future research.
> > >
> > > We have updated our draft to clarify these points and appreciate the reviewer’s effort once again.

---

### Official Review · Reviewer_nd8T · 2024-11-07

**Soundness:** 4
**Presentation:** 4
**Contribution:** 3
**Rating:** 8
**Confidence:** 4

**Summary:**

The authors derive a novel theoretical framework for off policy contextual bandits that strictly generalizes the complementary approaches of policy based and value based methods via a two stage decomposition involving both. It is argued that this provides a non-trivial advantage in large and structured action spaces. The two stage decomposition involves first using a policy based parametrization to identify an action cluster, followed by a regression based approach within each cluster to discriminate between the fine grained individual actions.

**Strengths:**

- Novel general approach for combining policy and value based learning methods in a hierarchical fashion so as to avoid bias and variance simultaneously.
- The paper is very well written and a pleasure to read
- Good discussion of related literature and its context for the current work.

**Weaknesses:**

- A good bit of the technical novelty in the current paper is shared with  prior work [Saito et. al. 2023] which the authors duly point out. Having said, that the prior work applies a similar idea for off policy estimation while the current work deals with policy optimization.

- No satisfactory general recipe to perform the right kind of clustering that can best exploit such a two stage decomposition.

- The main issue with a regression based approach is non-zero bias due to model capacity as the authors point out,  and in practice this error seems more plausible to have its source in an inability to fit within cluster, if at all, rather than across clusters. However, this is precisely what is assumed for a good performance in the second stage. If it is assumed to do sufficiently well in the second stage, wouldn't this somehow make the problem more likely to achieve a low error for a simple supervised learning global fit?

**Questions:**

- Clustering the action space based on the scalar embedding of averaged marginal rewards seems somewhat surprising and not justified even informally -- can you comment on this? Have you considered a random bucketing strategy as a baseline for the first stage, and if so, how much better does proposed method do over this baseline?


- Is there a typo in Equation (3)? -- should the third variance term on RHS be $\hat{q} (x, a) s_\theta (x, a)$ ? For easier verification, please consider adding a quick derivation of this to the appendix.


- Is there a fundamental reason for the theory to only work with the policy at the base stage and value at the second stage? A footnote on page 4 is discussing this, but the implication is unclear for whether this is a fundamental difference. The point about the importance weights being an issue doesn't seem convincing since the same reasoning applies to the first stage in the current proposal.

---

> ### Author Response · Authors · 2024-11-16
>
> We appreciate the valuable and thoughtful feedback from the reviewer. We respond to the concrete questions and comments in detail below.
>
> > Clustering the action space based on the scalar embedding of averaged marginal rewards seems somewhat surprising and not justified even informally -- can you comment on this? Have you considered a random bucketing strategy as a baseline for the first stage, and if so, how much better does proposed method do over this baseline?
>
> This is an interesting point to discuss. The intuition behind the heuristic clustering used in our experiments is that by grouping (marginally) better and worse actions separately, the first-stage policy effectively removes most of the worse actions (note that the first-stage policy is context-dependent, so the cluster with the highest marginal reward may not necessarily be the best cluster for some contexts). It is important to remove those actions in the first stage, because the potential overestimation of worse actions by the regression model $\hat{q}(x,a)$ is often the main source of performance degradation in the second stage. This cannot be achieved by merely performing random clustering.
>
> It is important to note that we acknowledge the current clustering method is heuristic. Indeed, in the synthetic experiment, POTEC with true clusters performed better than that with learned clusters, demonstrating the potential of POTEC with improved clustering methods.  As discussed in Section 5, we believe that developing a more principled approach for action clustering is possible and is an intriguing future direction.
>
> Following the reviewer’s interesting suggestion, we additionally compared POTEC with random clusters on synthetic data. The result is summarized in the following table, suggesting that heuristically learned clusters at least empirically outperform random clusters for a range of training data sizes, which is reasonable as discussed above.
>
> | training data sizes   |    500 | 1000 | 2000 | 4000 | 8000|
> |:--------------------------|--------:|--------:|--------:|--------:|--------:|
> | POTEC w/ **true clusters**   | 1.130 | 1.114 | 1.237 | 1.275 | 1.339 |
> | POTEC  w/ **learned clusters**  | 1.105 | 1.096  | 1.197  | 1.213 | 1.269 |
> | POTEC  w/ **random clusters** | 0.977 | 0.993 | 1.083 | 1.097 | 1.157 |
> | DR-PG (best among the baselines) | 0.903 | 0.910  | 0.997 | 1.096 | 1.140 |
>
> (Note that the values indicate the test policy value relative to that of the logging policy, i.e., $V(\pi_{\theta})/V(\pi_0)$.)
> > Is there a typo in Equation (3)? should the third variance term on RHS be $\hat{q}(x,a)s_{\theta}(x,a)$
>
> **Thank you for pointing this out, but that is NOT a typo**. We will provide the derivation in the revision, but as a quick reference, the following papers on off-policy **evaluation** (not **learning**, as in our work) derive the variance of the DR estimator. It can be observed that the term corresponding to the last term in our Eq. (3) depends on the true reward function, $q(x,a)$, rather than its estimate, $\hat{q}(x,a)$.
>
> - **Eq.(6)** of Yu-Xiang Wang, Alekh Agarwal, and Miroslav Dudık. Optimal and adaptive off-policy evaluation in contextual bandits. ICML2017.
> - **Lemma 5.2** of Audrey Huang, Liu Leqi, Zachary C. Lipton, and Kamyar Azizzadenesheli. Off-Policy Risk Assessment in Contextual Bandits. NeurIPS2021.
>
> >  Is there a fundamental reason for the theory to only work with the policy at the base stage and value at the second stage? A footnote on page 4 is discussing this, but the implication is unclear for whether this is a fundamental difference...
>
> The point about importance weights made in the footnote on page 4 is indeed the fundamental reason, and we will clarify it further here. First, it is important to understand that the POTEC gradient estimator for the first-stage policy relies on **cluster** importance weighting, not **vanilla** importance weighting as used in existing methods. This is the primary strength of our proposed gradient estimator and results in variance reduction shown in Proposition 3.5. This variance reduction is particularly advantageous for large action spaces, where vanilla importance weighting of existing methods produces extreme variance.
>
> The footnote argues that we lose this crucial advantage of the two-stage OPL framework if we use regression for the first stage and importance weighting for the second stage, because in this case, **vanilla** importance weighting is required to make the second-stage policy gradient unbiased. Consequently, the two-stage method would suffer from the same variance issue as IPS-PG and DR-PG.
>
> To fully grasp this point, it would be important to understand the distinction between cluster and vanilla importance weighting, as well as the role of cluster importance weighting to reduce variance in our method. We are more than happy to discuss this point further if our response above does not fully address the reviewer’s question.

---

> > ### Author Response · Authors · 2024-11-17
> > **Official Comment by Authors (Cont'd)**
> >
> > We would also like to address the following comment in the “weakness” section.
> >
> > >  The main issue with a regression based approach is non-zero bias due to model capacity as the authors point out, and in practice this error seems more plausible to have its source in an inability to fit within cluster, if at all, rather than across clusters. However, this is precisely what is assumed for a good performance in the second stage. If it is assumed to do sufficiently well in the second stage, wouldn't this somehow make the problem more likely to achieve a low error for a simple supervised learning global fit?
> >
> > This is a very interesting question. We believe it is important to understand that achieving globally optimal decision-making is a sufficient condition for locally optimal decision-making within clusters (a globally optimal model is locally optimal, but a locally optimal model is not necessarily globally optimal). Therefore, the second stage of POTEC indeed focuses on a simpler problem compared to the regression-based method.
> >
> > This is why the regression-based method never outperformed POTEC in either synthetic or real-world data in our experiments. Even if we hypothetically consider the case where the regression-based method may perform better as the reviewer suggested, POTEC does not underperform the regression-based baseline with a reasonable choice of the number of clusters ($|\mathcal{C}|$). This is because POTEC generalizes the policy-based and regression-based approaches as described in Section 3.3, and by setting the number of clusters to one, POTEC reduces to the regression-based method and, therefore, performs at least equally by construction.

---

### Official Review · Reviewer_J9Lw · 2024-11-10

**Soundness:** 4
**Presentation:** 4
**Contribution:** 3
**Rating:** 8
**Confidence:** 3

**Summary:**

This paper introduces POTEC, a novel off-policy learning (OPL) method designed to handle large action spaces, a common challenge in real-world recommendation systems. The algorithm decomposes the policy into two stages: a first-stage policy that selects promising action clusters, and a second-stage policy that chooses specific actions within the selected cluster. The paper provides theoretical analysis showing that POTEC achieves unbiased gradient estimation under a local correctness condition. It also shows its cluster-based importance weighting leads to substantially lower variance compared to traditional methods. The authors present a complete learning framework, including how to optimize both policies through a combination of policy-based and regression-based approaches. Through extensive experiments on both synthetic and real-world recommendation data, POTEC demonstrates superior performance compared to existing methods such as IPS-PG and DR-PG, particularly when dealing with sparse action observations in logged data and large action spaces.

**Strengths:**

The paper introduces a novel two-stage approach to address the challenges of large action spaces in off-policy learning.

It demonstrates high quality through rigorous theoretical proof of the unbiasedness of the POTEC gradient estimator under local correctness condition, and shows how cluster-level importance weighting effectively reduces variance.

The paper provides clear theoretical justification for the two-stage decomposition and progresses logically from the problem formulation to the solution.

the paper addresses fundamental challenges in large-scale off-policy learning by providing a unified framework with theoretical guarantees for the joint optimization of two-stage policies.

**Weaknesses:**

The paper did not specify how different clustering algorithm could impact the performance of POTEC from both theoretical point of view and empirical study.

There could be more details provided on how the test data is constructed in empirical study.

**Questions:**

How test data is constructed in empirical study (both synthetic and real world experiment)? For example, is there overlap of context between test data and training data, if so how much overlap?

What will happen if full cluster support condition is not met in logging policy? In the real world, how do you check if this condition is met?

---

> ### Author Response · Authors · 2024-11-16
>
> We appreciate the valuable and thoughtful feedback from the reviewer. We respond to the concrete questions and comments in detail below.
>
> > The paper did not specify how different clustering algorithm could impact the performance of POTEC from both theoretical point of view and empirical study.
>
> This is a great point. From a theoretical standpoint, our current analysis already provides valuable insights. Specifically, the bias analysis in Theorem 3.2 indicates that the bias in the first-stage policy gradient estimation decreases when the estimation error for the relative value differences, $\Delta_{q}(x,a,b)$, within each cluster is small. Therefore, an effective clustering algorithm should lead to a better estimation of relative value differences within each cluster by a given regression model $\hat{q}(x,a)$.
>
> In the synthetic experiment, we have already compared POTEC with true clusters and with learned clusters, demonstrating the impact of clustering quality on POTEC’s performance and the potential of POTEC with improved clustering methods.
>
> Another reviewer was interested in comparing POTEC with random clusters, and we performed such an additional ablution study on synthetic data. The result is summarized in the following table, suggesting that heuristically learned clusters empirically outperform random clusters for a range of training data sizes.
>
> | training data sizes  |    500 | 1000 | 2000 | 4000 | 8000|
> |:--------------------------|--------:|--------:|--------:|--------:|--------:|
> | POTEC w/ **true clusters**   | 1.130 | 1.114 | 1.237 | 1.275 | 1.339 |
> | POTEC  w/ **learned clusters**  | 1.105 | 1.096  | 1.197  | 1.213 | 1.269 |
> | POTEC  w/ **random clusters** | 0.977 | 0.993 | 1.083 | 1.097 | 1.157 |
> | DR-PG (best among the baselines) | 0.903 | 0.910  | 0.997 | 1.096 | 1.140 |
>
> (Note that the values indicate the test policy value relative to that of the logging policy, i.e., $V(\pi_{\theta})/V(\pi_0)$.)
>
> > How test data is constructed in empirical study (both synthetic and real world experiment)? For example, is there overlap of context between test data and training data, if so how much overlap?
>
> We are happy to clarify this point. In the synthetic experiment, both the training and testing data follow the same distribution (Gaussian), so there is no covariate shift and their context spaces completely overlap. Please note that we ensured the training and testing data are different samples to evaluate and compare the generalization performance of the OPL methods.
>
> For the real-world experiment, we first randomly split the users into training (50%) and test (50%) groups. Then, we draw $n$ users ($x_i$) from the training set with replacement. For each context, we sample action $a_i$ from the logging policy and then sample the reward $r_i$ given $(x_i, a_i)$. This constructs the training logged dataset of the form $\mathcal{D} = \\{(x_i, a_i, r_i)\\}_{i=1}^n$. After training the policies, we evaluate their performance on the test set using the true reward function $q(x, a)$. This process is repeated 30 times with different draws in the training group. Since the user split and data sampling are random, the context distributions in the training and testing sets are, in expectation, identical. We will clarify these settings in the revision.
>
> > What will happen if full cluster support condition is not met in logging policy? In the real world, how do you check if this condition is met?
>
> We are happy to clarify this as well. If the full cluster support condition is not met, the proposed estimator for the policy gradient in Eq.(7) will be biased. However, the full cluster support is weaker than the standard cluster support (Condition 2.1), and thus existing estimators would produce larger bias in this case.
>
> It is straightforward to analyze the bias of the POTEC gradient estimator when the full cluster support condition is violated by following the relevant derivation in (Sachdeva et al. 2020). Similarly, it is straightforward to verify whether this condition is satisfied by following its definition. Specifically, we first calculate the marginalized logging distribution regarding the cluster space, i.e., $\pi_0(c|x) = \sum_a \mathbb{I} \\{c_a = c \\} \pi_0(a|x)$, and then check if the condition $\pi_0(c|x) > 0$ holds for all $c$ and $x$.
>
> - (Sachdeva et al. 2020) Noveen Sachdeva, Yi Su, and Thorsten Joachims. Off-policy bandits with deficient support. KDD2020.

---

> > ### Comment · Reviewer_J9Lw · 2024-11-27
> >
> > Thank you for providing the additional analysis on the impact of the clustering method using random clusters, as well as for clarifying the empirical study's sampling process and explaining how to verify if the full cluster support condition is met. I maintain my score of 8.

---

> > > ### Author Response · Authors · 2024-11-27
> > >
> > > We would like to thank the reviewer for their response. If the reviewer has any remaining questions or comments about our work, we would be more than happy to discuss them further.

---

### Meta-Review · Area_Chair_hwA5 · 2024-12-21

**Metareview:**

This paper studies a novel two-stage off-policy learning algorithm, POTEC, designed for large discrete action spaces by combining policy- and regression-based approaches via clustering and policy decomposition. The method provides unbiased gradient estimation under local correctness conditions and significantly reduces variance through cluster-level importance weighting. Empirical results demonstrate substantial improvements over existing methods across synthetic and real-world datasets. While reviewers praised the theoretical soundness, clarity, and strong empirical performance, they raised concerns about the heuristic clustering approach, limited discussion of cluster selection, and lack of experiments with varying logging policies. Despite these minor issues, the reviewers were positive overall, recommending acceptance.

**Additional Comments On Reviewer Discussion:**

During the rebuttal period, reviewers raised several points: clarification on the clustering algorithm's impact and selection of cluster count, handling of unknown logging policies, robustness to general reward and logging policy structures, and additional empirical details, particularly about real-world data and context-dependent clustering. The authors provided detailed responses, including a discussion of heuristic clustering's theoretical and empirical effects, recommendations for hyperparameter tuning via off-policy evaluation, and clarification on the method's robustness to different reward structures. They also highlighted potential for future research on clustering methods and included additional experiments comparing random and learned clusters. Reviewers acknowledged these clarifications, maintained high scores, and suggested adding more experiments on varying logging policies. These thorough responses addressed the key concerns, and the constructive exchange demonstrated the paper's strengths, contributing to the final decision to recommend acceptance.

---

### Decision · Program_Chairs · 2025-01-22

Accept (Spotlight)